# Aneuploidy-induced proteostasis disruption impairs mitochondrial functions and mediates aggregation of mitochondrial precursor proteins through SQSTM1/p62

Prince Saforo Amponsah [1] ✉, Jan-Eric Bökenkamp [1], Olha Kurpa [1], Svenja Lenhard[2], Anna Myronova[1], Daniel Osmar Vega Velazquez[1], Celina Hirschelmann[1], Christian Behrends [3], Johannes M. Herrmann [2], Markus Räschle [1] & Zuzana Storchová [1] ✉

Aneuploidy, or aberrant chromosomal content, disrupts cellular proteostasis through altered expression of numerous proteins. Aneuploid cells accumulate SQSTM1/p62-positive cytosolic bodies, exhibit impaired protein folding, and show altered proteasomal and lysosomal activity. Here, we employ p62 proximity- and affinity-based proteomics to elucidate p62 interactors in aneuploid cells and observe an enrichment of mitochondrial proteins. Increased protein aggregation and colocalization of p62 with both novel interactors and mitochondrial proteins is further confirmed by microscopy. Compared to parental diploids, aneuploid cells suffer from mitochondrial defects, including perinuclearly-clustered mitochondrial networks, elevated reactive oxygen species levels, reduced mitochondrial DNA abundance, and impaired protein import, leading to cytosolic accumulation of mitochondrial precursor proteins. Overexpression of heat shock proteins in aneuploid cells mitigates protein aggregation and decreases the colocalization of p62 with the mitochondrial protein TOMM20. Thus, proteotoxic stress caused by chromosome gains results in the sequestration of mitochondrial precursor proteins into cytosolic p62-bodies, thereby compromising mitochondrial function.

While most healthy human cells maintain a diploid chromosome content, approximately 90% of solid tumors contain cells with aberrant chromosome numbers, known as aneuploidy[1]. This is often associated with a high rate of chromosome segregation errors, referred to as chromosomal instability (CIN). The persistent occurrence of chromosome gains and losses results in the generation of novel karyotypic variations[2]. Analyses of large cancer datasets as well as functional analyses of aneuploid cells elucidated the pivotal role of

aneuploidy in tumorigenesis; in promotion of tumor growth, metastasis, and the development of drug resistance[3,4].

Yet, aneuploidy comes at a cost. The gain or loss of even a single chromosome is generally incompatible with the viability of human embryos. The few exceptions to this rule, such as trisomy of chromosomes 13, 18, or 21, are often accompanied by a multitude of pathologies. Lab-engineered human constitutively aneuploid cells manifest numerous defects. Cells lacking a chromosome (monosomy)

[1]Molecular Genetics, Rheinland-Pfälzische Technische Universität (RPTU) Kaiserslautern-Landau, Kaiserslautern, Germany. [2]Cell Biology, Rheinland-Pfälzische Technische Universität (RPTU) Kaiserslautern-Landau, Kaiserslautern, Germany. [3]Munich Cluster for Systems Neurology (SyNergy), Faculty of Medicine, Ludwig-Maximilians-Universität München, Munich, Germany. ✉e-mail: amponsah@rptu.de; zuzana.storchova@rptu.de

show reduced proliferation, reduced translation and increased DNA damage, and their viability is incompatible with the presence of functional *TP53*[5–7]. Human cells with extra chromosomes (polysomy) also frequently exhibit slower proliferation rates when compared to their diploid counterparts. Moreover, they manifest an array of phenotypic changes, including features of proteotoxic stress, as evidenced by an increased sensitivity to inhibitors targeting protein folding and turnover, elevated replication stress and accumulation of DNA damage and structural rearrangements, activation of the innate immune response and reduced translation[5,8–13]. Similar effects were also observed in acute response to aneuploidy, immediately after induced chromosome missegregation[14–16].

The primary cause of the stresses associated with chromosome gains is largely considered to be the increased expression of several hundreds or thousands of surplus proteins encoded on the extra chromosomes[8,17,18]. These additional proteins overwhelm the machinery dedicated to protein synthesis, folding, and degradation, leading to proteotoxic stress. Indeed, aneuploid cells with supernumerary chromosomes often activate integrated stress response, accumulate protein aggregates, and exhibit defects in protein folding as well as increased autophagy and proteasomal activity[9,10,14,18]. Oxidative stress and elevated production of reactive oxygen species have also been observed in both immediate and late responses to chromosome missegregation. Among the most striking features observed in aneuploid cells is the accumulation of cytosolic deposits positive for sequestosome 1 (SQSTM1/p62), an autophagy receptor and ubiquitin-binding protein that sequesters misfolded proteins, protein aggregates and damaged organelles, and targets them for degradation. Cytosolic p62-bodies accumulate in aneuploid cells immediately after induced chromosome missegregation as well as in cells with constitutive chromosome gain[8,14–16,19]. The causes of p62 accumulation remain only partially understood. In acute response to chromosome missegregation, p62, which is an autophagy substrate, accumulates due to saturated autophagy[14,15]. In cells with constitutive chromosome gain, the increased expression of p62 is observed on both transcriptome and proteome levels[8,16,19], likely in response to the excess or misfolded proteins, or in response to increased oxidative stress. Taken together, accumulation of cytosolic p62 bodies is integral to the cellular response to aneuploidy, although their function remains unclear.

p62 is a versatile protein with crucial functions in cellular protein homeostasis through both autophagy and the proteasome. It acts as a receptor in selective autophagy, where it recognizes ubiquitinated cargo and interacts with autophagy-related proteins, such as LC3, to facilitate cargo engulfment into autophagosomes[20–22]. Moreover, due to its ability to recognize and interact with ubiquitin, p62 connects ubiquitinated proteins to the autophagic machinery, facilitating their degradation through selective autophagy[23–25]. p62 also interacts with numerous signaling molecules, including kinases and transcription factors, regulating cellular processes such as oxidative stress response, inflammation, and cell survival[22,26]. Additionally, p62 acts as a key mediator in mitochondrial turnover, where it functions as an adaptor protein for PINK and Parkin mediated mitophagy[27]. Although p62 primarily functions in autophagy, it also interacts with the proteasomal degradation pathway. p62 captures polyubiquitinated substrates through its UBA domain and recruits them to the 26S proteasome. For example, by interacting with KEAP1, it regulates NRF2 degradation via the proteasome, thereby controlling cellular antioxidant responses. Similarly, it interacts with RBX1, a key component of E3 ubiquitin ligase complexes, in PINK and Parkin independent degradation of mitochondrial proteins[28,29]. Interestingly, p62-containing aggregates have been detected in extracellular vesicles in cancer cells and in neurodegeneration, suggesting its role in unconventional secretion[30]. Dysregulation of p62 has been implicated in various pathological conditions, including neurodegenerative disorders and cancer, highlighting its importance in protein aggregation and clearance[31–33].

However, it remains unclear which of the many functions of p62 is required in response to chromosome gain.

Here, we used our previously established model system of human cell lines engineered to constitutively carry one or two extra chromosomal copies (trisomy and tetrasomy, hereafter polysomy[8,34]) to analyze the function of p62 in response to aneuploidy. We found that the abundance of p62 scales with the number of aneuploid chromosomes in model systems as well as in cancer cell lines of the Cancer Cell Line Encyclopedia (CCLE) based on the analysis of multi-omics data retrieved from the Dependency Map (DepMap) data portal. Labeling of p62-proximal proteins via APEX2-p62-dependent biotinylation as well as immunoprecipitation of p62 interactors, followed by mass spectrometry, revealed that a large fraction of p62 interactors in aneuploid cells comprises of mitochondrial proteins. These findings were further corroborated with increased colocalization of p62 with mitochondrial markers. We show that the sequestration of mitochondrial proteins into p62-bodies is enhanced in aneuploid cells and can be rescued by improving cytosolic protein homeostasis through expression of protein folding factors. Finally, we demonstrate that proteotoxic stress due to gain of a single chromosome impairs mitochondrial precursor protein import, which leads to their sequestration into p62-positive cytosolic bodies.

## Results

### Abundance of p62 in aneuploid cells scales with the amount of extra DNA

We and others have previously shown that aneuploid cells accumulate autophagy and lysosomal markers on both transcriptome and proteome levels[8,14–16,19]. Among the most abundant proteins within this category was the selective autophagy receptor p62, which forms highly abundant cytosolic p62-bodies in polysomic cells. To evaluate the composition of the p62-bodies, we used engineered cell lines with extra chromosomes 3, 5, 13 and 21 transferred into pseudodiploid HCT116 cell line via microcell mediated chromosome transfer (Fig. 1a). The karyotypes were confirmed by chromosome painting and whole genome sequencing[8,34], and the increased protein expression from the extra chromosomes was verified by mass spectrometry ([8], Supplementary Fig. 1a). The cell lines were labeled according to their karyotype as HCT116 3/3 (trisomy of chromosome 3), 5/3 (trisomy of chromosome 5), 5/4 (tetrasomy of chromosome 5), 13/3 (trisomy of chromosome 13) and 21/3 (trisomy of chromosome 21). Immunoblotting of cell lysates from these five polysomic cell lines, compared to the parental diploid, revealed that the expression of p62 increases with the increased size of the extra chromosome, hence with the increasing number of additional protein-coding genes (Fig. 1b, c). The p62 abundance further increased upon treatment with an inhibitor of autophagosome-lysosome fusion Bafilomycin A1 (BafA1), confirming normal autophagic flux in the polysomic cells, as previously reported[8,19] (Fig. 1b, c). The size and number of p62 bodies per cell surface area also increase in polysomic cells (Fig. 1d–f). It should be noted that *SQSTM1* is located on chromosome 5 and thus its increased expression is expected in trisomy and tetrasomy of chromosome 5. However, the p62 abundance increased in all polysomic cells and the number and size of the bodies tightly correlated with the amount of surplus protein coding genes irrespective of the identity of the extra chromosome (compare trisomy of chromosome 5 and trisomy of chromosome 3, Fig. 1g). The polysomic cells also showed increased staining with a red fluorescent molecular rotor dye, which stains protein aggregates (Supplementary Fig. 2a-b). The signals correlated with the number of surplus proteins and were often positive for p62 (Supplementary Fig. 2a, c). Protein aggregates were previously shown to accumulate in response to aneuploidy in both yeast and mammalian cells[8,18,35]. Taken together, the increase in p62 foci number and size in polysomic cells is consistent with increased protein aggregation.

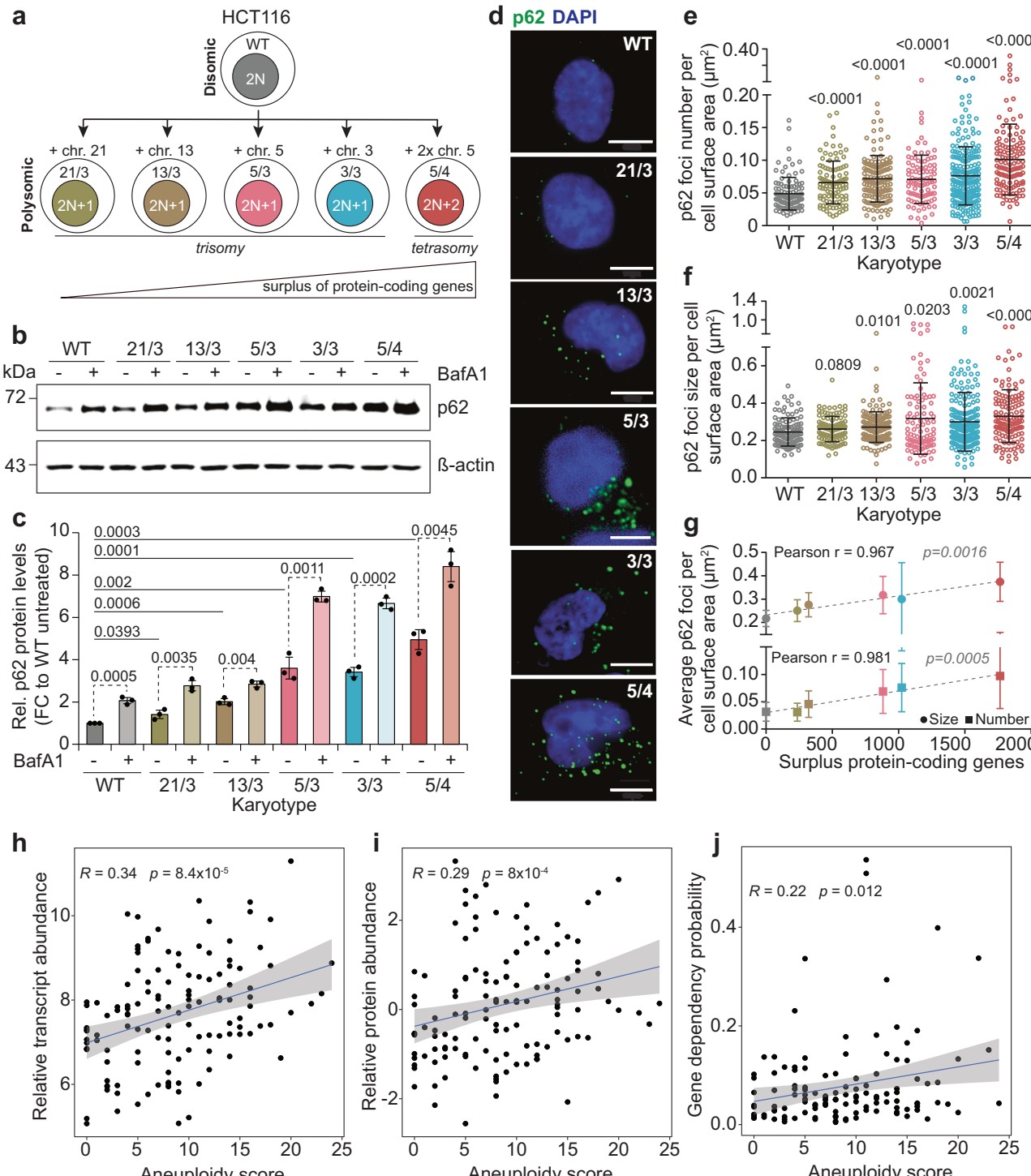

**Fig. 1 | Expression levels and cytosolic deposits of p62 are increased in polysomic cells and correlate with aneuploidy. a** Schematic depiction of the cell lines used. **b** Representative immunoblot of p62 without and with Bafilomycin A1 (BafA1) treatment. ß-actin serves as a loading control. **c** Quantification of p62 expression levels without and with BafA1 treatment. Data is shown as mean ± s.d. fold change to WT from *n* = 3 independent experiments, and individual replicates are plotted as dots. *P* values represent two-tailed unpaired Student's *t*-test. **d** Representative confocal images of p62 foci immunofluorescence (green). DAPI (blue) stains DNA (nuclei). Images are collapsed from Z-stacks. Scale bar is 10 μm. Quantification of (**e**) p62 foci number per cell surface area in μm², and **f** p62 foci size per cell surface area in μm². Individual data for each cell from *n* = 3 independent experiments and mean ± s.d. are shown. *P* values represent non-parametric ANOVA (Kruskal–Wallis statistic for (**e**) 99.54, *p* < 0.0001; for (**f**) 30.37, *p* < 0.0001) followed by Dunn's multiple comparisons test. **g** Correlations of the mean ± s.d. p62 foci number and size with the number of surplus protein-coding genes in all the cell lines. Two-tailed *P* values are indicated. Correlations of (**h**) relative transcript abundance, **i** relative protein abundance, and **j** gene dependency probability, with aneuploidy score of 127 cancer cell lines from the Cancer Cell Line Encyclopedia (CCLE). *R* represents Spearman's rank correlation coefficient. *P* values are two-sided.

The correlation between aneuploidy and p62 abundance is not restricted to our model polysomic cell lines. Integration of the genomic[36,37], transcriptomic[38], and proteomic[39] data obtained for cancer cell lines of the CCLE and stored in the DepMap database[40] showed that both the transcript and protein abundance of p62 increase with aneuploidy score, which is defined as the number of chromosome arm-level aberrations[41] (Fig. 1h, i). Moreover, the probability of gene dependency for *SQSTM1*, which estimates the reliance of cancer cells on a particular gene, also correlates with the aneuploidy score of cancer cells (Fig. 1j). Together, these data establish that p62 abundance increases with increasing degree of aneuploidy in cancer cells.

## p62-dependent cytosolic proximity labeling reveals juxtaposition with mitochondrial proteins

To assess the function of p62 in polysomic cells, we decided to identify proteins associated with the cytosolic p62-bodies. To this end, we analyzed p62-proximal proteins by performing APEX2-p62 based proximity biotinylation proteomics[42]. We stably expressed *myc-APEX2-SQSTM1* construct in the parental diploid, 13/3, 3/3, and 5/4 cells by lentiviral transduction (Supplementary Fig. 3a). Transduced cells were then treated with biotin-phenol for 30 min and a pulse (1 min) of 1 mM $H_2O_2$ to induce the biotinylation reaction. Biotinylated proteins were enriched by streptavidin pull down and analyzed by mass spectrometry (Fig. 2a). By this approach, we identified 2321-3302 proteins biotinylated in the proximity of p62 in the cytosol (Supplementary Fig. 1b). Principal component analysis showed clustering of quadruplicate measurements across cell lines (Supplementary Fig. 1c). To adjust for differences in total protein abundance between the cell lines, we analyzed global protein abundance by separate quantitative proteomics experiments. The fold changes in p62-proximal proteins (hereafter called proxitome) in the polysomic cells relative to the parental diploid were then statistically tested against the corresponding fold changes in global protein abundance. From this analysis, we identified 236-902 proteins significantly increased in the p62 proxitome of 13/3, 3/3, and 5/4 cell lines (Fig. 2b, and Supplementary Data 1). Cellular compartment over-representation analysis showed that mitochondrial proteins were the predominant candidates within the polysomy-specific proxitome of p62 (Fig. 2c, Supplementary Fig. 3b, and Supplementary Data 1). The fraction of mitochondrial proteins in the p62-proxitome was significantly higher than their fraction in the total proteome of these cells (Fig. 2d). Meanwhile, there was no uniform global expression changes of mitochondrial proteins within the polysomic cell lines when normalized to the diploid parental cell line (i.e., lower in 13/3, higher in 3/3, no difference in 5/4, Supplementary Fig. 1d). Additionally, proteins associated with chaperonin containing T-complex were enriched in 3/3 and 5/4 cells, external encapsulating structure organization as well as extracellular matrix were enriched in both 13/3 and 5/4 cells, and cytosolic ribosomal subunits were enriched in the 3/3 cell line (Fig. 2c). Strikingly, the p62 proxitome was not significantly enriched for proteins encoded on the extra chromosomes. Thus, proximity labeling shows that the cytosolic p62 interactome in cells with extra chromosomes is enriched with mitochondrial proteins.

To analyze the p62 proximal mitochondrial proteins in detail, we evaluated their distribution by sub-compartmental localization. By using the TargetP 2.0 prediction tool[43], we found a significantly increased enrichment of proteins that are predicted to possess matrix targeting sequences (MTS) in the 13/3 and 3/3 cell line, while there appears to be equal distribution of MTS and non-MTS containing proteins in the 5/4 cell line (Supplementary Fig. 3c). Proteins of the inner membrane (IM) and mitochondrial matrix were enriched in the polysomy-specific p62 proxitome, and outer mitochondrial membrane (OM) proteins were additionally enriched in the p62 proxitome of the 5/4 cells (Supplementary Fig. 3d). The p62 proxitome in the 3/3 and 5/4

cell line was also significantly enriched with hydrophobic proteins, while no enrichment for subunits of macromolecular complexes was identified (Supplementary Fig. 3e, f). Taken together, our data indicate that p62 bodies in polysomic cells are enriched for inner membrane and mitochondrial matrix proteins with MTS, which is suggestive of impaired mitochondrial import and function.

## Spatially restricted p62-dependent proximity labeling does not show enrichment for mitochondrial proteins in autophagosomes

The autophagy receptor p62 is known for its role in targeting selected cargo to autophagosomes. To elucidate the p62-interactome channeled through autophagy, we profiled the content of autophagosomes by proximity labeling using APEX2-p62 and limited proteolysis[42]. Confocal microscopy of the transduced cell lines revealed colocalization of the myc-APEX2-p62 bait and biotin-positive puncta, which in turn colocalized with the lysosomal marker LAMP1 upon BafA1 treatment and proximity labeling (Supplementary Fig. 4a). Immunoblotting confirmed that the APEX2 fusions, biotinylated proteins, and endogenous p62 were partly protected from proteinase K, and enriched for lipidated LC3B-II form (Supplementary Fig. 4b). These results indicate that the APEX2-p62 chimeras are targeted to autophagosomes and biotinylate engulfed proteins, as previously shown[42]. To perform the analysis, we treated the transduced cells with biotin-phenol (30 min) and subsequent pulsing with $H_2O_2$ (1 min). Clarified lysates were incubated with proteinase K (PK) to digest proteins that were not protected by intact membranes (Fig. 3a).

MS was conducted after lysis of the enriched autophagosomes and pulldown of the biotinylated proteins using streptavidin agarose, in both BafA1 treated and untreated cells (Fig. 3a). We included a PK resistant control, in which clarified lysates were treated with both PK and RAPIGest to identify proteins not digested by PK. We detected 165 PK-resistant proteins, which were excluded from our analyses (Supplementary Fig. 1e). In total, we identified 80−935 and 536−1107 p62-proximal proteins enriched in the autophagosome lumen of the cell lines treated with DMSO and BafA1, respectively (Supplementary Fig. 1e). Principal component analysis showed clustering of triplicate measurements across cell lines and treatment conditions (Supplementary Fig. 1f). First, we asked what proteins were enriched among p62 proximal proteins in the autophagosome lumen of polysomic cells compared to parental cells in normal conditions (no Baf1A treatment). To this end, we tested the fold changes in p62 proxitome in autophagosomal lumen of the polysomic cells normalized to parental diploid against the corresponding fold changes in global protein abundance (Supplementary Fig. 5a) and identified 15-156 significantly enriched proteins in the 13/3, 3/3 and 5/4 cell lines (Supplementary Fig. 5a, and Supplementary Data 2). While this included several mitochondrial proteins, there was no significant over-representation of mitochondrial proteins. Instead, cytosolic ribosomes and ribonucleoproteins were the predominant candidates over-represented within the polysomy-specific autophagosome lumen p62 proxitome (Supplementary Fig. 5b, c, and Supplementary Data 2). Next, we assessed the statistical significance of differential protein abundance values between p62 proxitome in autophagosome lumen of BafA1-treated cells, normalized to untreated cells to derive a set of BafA1-stabilized p62 cargo proteins for each karyotype (Fig. 3b, Supplementary Fig. 4c, and Supplementary Data 2). We observed enrichment for hATG8s (e.g., GABARAPL2), selective autophagy receptors (e.g., CALCOCO1, CALCOCO2, NBR1), established p62 substrates (e.g., KEAP1), and several known p62 cargo proteins (e.g., TBK1). Autophagy proteins as well as catalogued p62-specific selective autophagy cargos[42] were significantly over-represented among the identified p62 cargo proteins in all cell lines independently of their karyotype (Fig. 3c, d), thus authenticating our assay. Furthermore, immunostaining of p62 and LC3B, a well characterized autophagy marker, did not show any

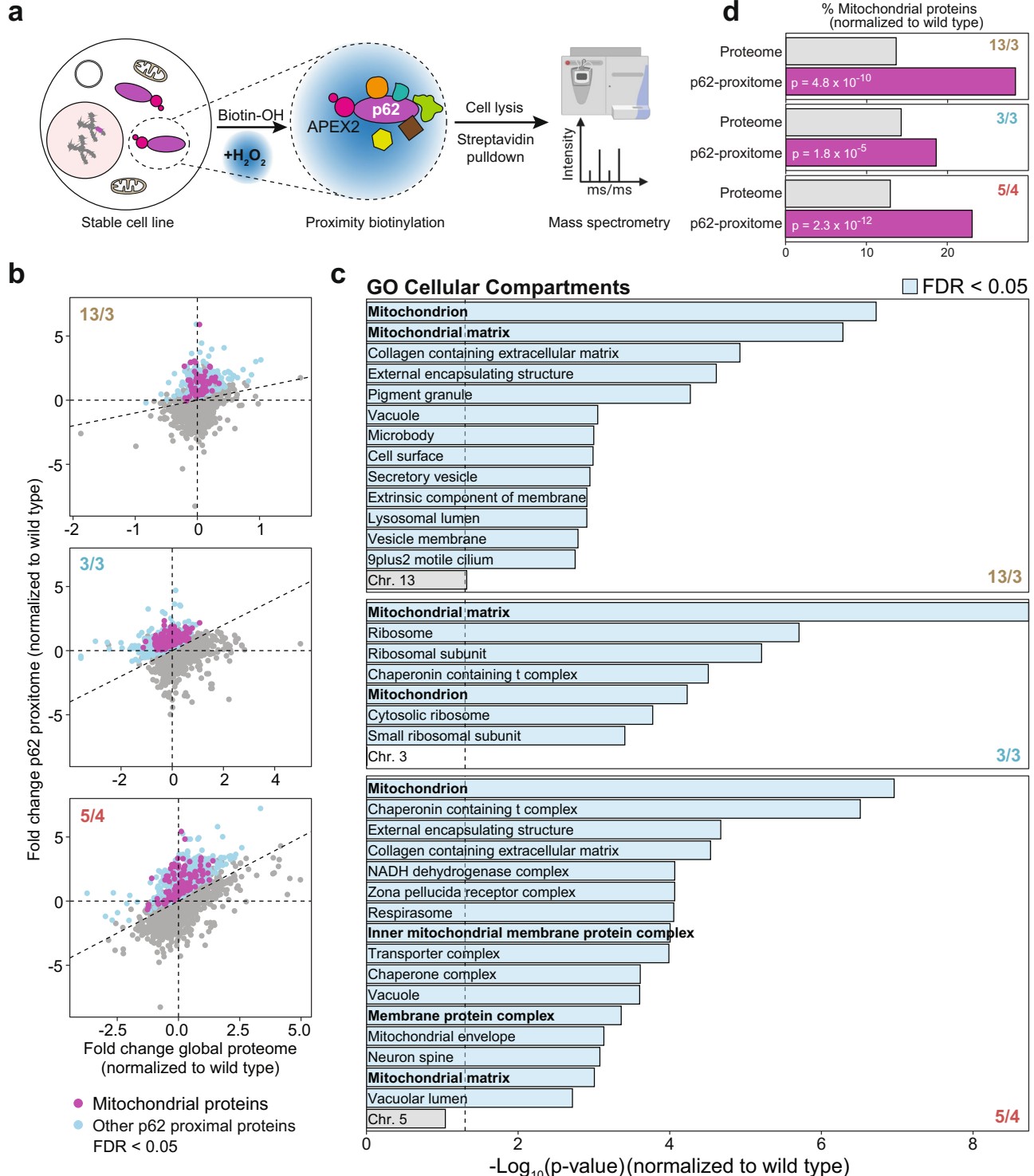

**Fig. 2 | APEX2-based proximity biotinylation of p62 interactors followed by label-free proteomics reveals enrichment of mitochondrial proteins in polysomic cells. a** Schematic depiction of the experimental procedure for mass spectrometry-based identification of p62-proximal proteins through APEX2-mediated proximity biotinylation. Image partly created in BioRender. **b** Scatter plot of the fold changes in global protein abundance in the 13/3, 3/3 and 5/4 polysomic cell lines relative to the parental cell line (WT) and the corresponding changes in abundance of cytosolic p62-proximal proteins relative to WT. Colored dots represent proteins with significantly higher abundance changes of p62-proximal proteins in polysomic cells (FDR < 0.05). Data is generated from mass spectrometry analysis of *n* = 4 biological replicates (see Supplementary Fig. 1b, c, and Supplementary Data 1). **c** Gene ontology (GO) over-representation analysis of the enriched p62-proximal proteins in (**b**). Blue bars represent negative log-transformed *p* values (one-sided hypergeometric test) for GO terms with FDR < 0.05. Representatives from clusters of related GO terms are shown (see "Methods"). **d** Percentage of mitochondrial proteins in the measured global proteome (grey) and the corresponding percentage within the p62 proximal proteome (magenta) of 13/3, 3/3 and 5/4 cells relative to WT. *P* values represent results of one-sided hypergeometric test, evaluating the differences in representation of mitochondrial proteins relative to the measured proteome.

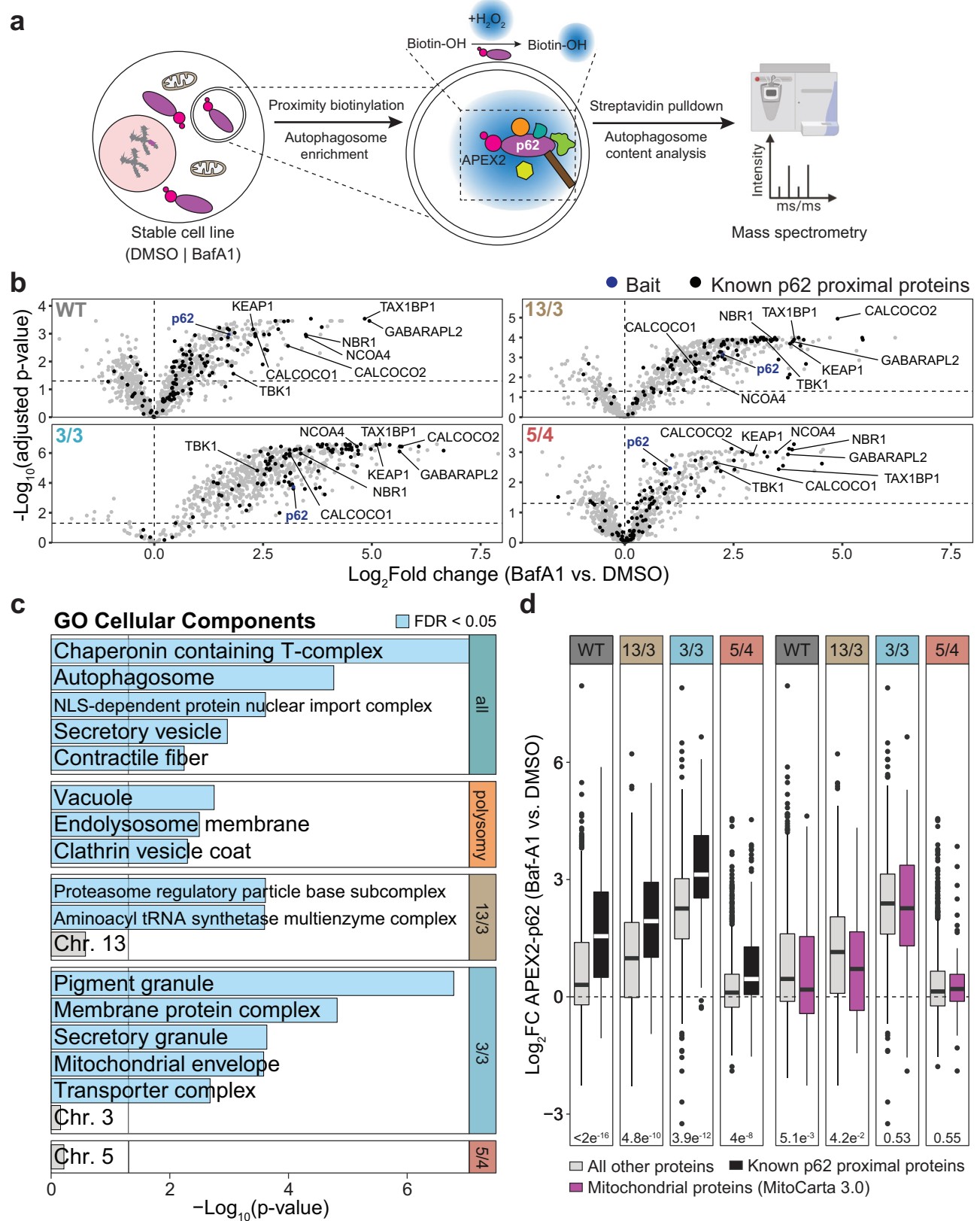

significant differences in the Pearson correlation coefficient of their colocalization in the diploid parental and polysomic cells (Supplementary Fig. 5d, e). Importantly, while cellular compartment overrepresentation analysis identified mitochondrial envelope proteins among the BafA1-stabilized p62 cargo in autophagosome lumen of the 3/3 cells, there was no significant over-representation of mitochondrial

proteins, as defined by the robust MitoCarta3.0 inventory[44] (Fig. 3c, d). There was also no significant over-representation of proteins encoded by genes on the extra chromosomes. Interestingly, the p62 cargo common to all polysomic cell lines, but not the parental diploid cells, included proteins mainly related to endocytic vesicle formation. Together, this suggests that while mitochondrial proteins are

**Fig. 3 | Mitochondrial proteins are not overrepresented p62 cargo candidates in autophagosome lumen. a** Schematic depiction of experimental procedure for autophagosome content profiling to identify p62 cargo candidates via APEX2-mediated proximity biotinylation. Image partly created in BioRender. **b** Volcano plots showing log$_2$-transformed fold change in BafA1 versus DMSO (vehicle) treated conditions across the cell lines used. Selected known p62 interactors are annotated. Data is generated from mass spectrometry analysis of $n$ = 3–4 biological replicates (see Supplementary Fig. 1e, f, and Supplementary Data 2). **c** Gene ontology (GO) over-representation analysis of the p62 cargo candidates common to all karyotypes, shared among all polysomic cell lines, and exclusive to each karyotype

(Supplementary Fig. 4c). Blue bars represent negative log-transformed $p$ values (one-sided hypergeometric test) for GO terms with FDR < 0.05. **d** Boxplots indicating fold changes in abundance of catalogued p62-specific autophagosome lumen cargo ([42], black) and mitochondrial proteins as defined by the MitoCarta3.0 inventory (magenta) compared to all other proteins (grey) enriched among the p62 cargo candidates from (**b**). Boxplots represent median with 25th and 75th percentile. Whiskers extend from upper and lower bound of the box to the largest and smallest values, respectively, no further than 1.5x inter-quartile range from the respective bound (Tukey method). *P* values are derived from two-sided Wilcoxon's ranks sum tests.

---

sequestered into p62-positive cytosolic bodies in polysomic cells, these proteins are not targeted by p62 into autophagosomes.

### Affinity proteomics confirm mitochondrial proteins as predominant cytosolic interactors of p62 in polysomic cells

To complement the p62-proximity proteomics experiments, we performed immunoprecipitation of p62 from whole-cell lysates of two polysomic cell lines, HCT116 13/3 and 5/4, and the parental cell line HCT116, followed by mass spectrometry analysis (IP-MS). We analyzed untreated cells, as well as cells treated with the autophagy inhibitor BafA1 to stabilize the p62 levels and thereby to enrich the fraction of potential interactors. The data was normalized to a pull-down performed in the same lysates using IgG as a bait (Fig. 4a). In total we found 435-941 proteins enriched with p62 from quadruplicate experiments (Supplementary Fig. 1g). Principal component analysis showed clustering of quadruplicate measurements across cell lines, treatment conditions and baits (Supplementary Fig. 1h). We assessed the statistical significance of differential protein abundance values between p62 pulldowns and the IgG controls to derive a set of candidate interactors of p62 for each karyotype and treatment condition (Supplementary Data 3). We identified NBR1, TAX1BP1, KEAP1 and other well-known interactors of p62 in all analyzed cell lines independently of their karyotype (Fig. 4b, and Supplementary Fig. 6a)[42]. Additionally, we identified candidates exclusive to both polysomic cell lines: the molecular chaperone HSPB1 (HSP27), lysosomal protease CTSZ, mRNA surveillance and ribosome quality control factor PELO, as well as the mitochondrial hydroxyacyl-CoA dehydrogenase trifunctional multienzyme complex subunits HADHA and HADHB (Fig. 4b, and Supplementary Fig. 6a). To validate the interactors, we evaluated their colocalization with p62 by immunofluorescence microscopy. This analysis confirmed colocalization of p62 with the previously identified interactors NBR1, TAX1BP1 and KEAP1 (Supplementary Fig. 6b). The newly identified and previously unknown polysomy-specific p62 interactors CTSZ and PELO also colocalized with p62 in the polysomic cells, but rarely in the parental cell line. Importantly, this colocalization was also observed in the 3/3 cell line, which was not subjected to IP-MS (Supplementary Fig. 6b).

Hierarchical clustering of enriched p62-interactors according to their z-scores revealed frequent mitochondrial proteins in polysomic cells (Supplementary Fig. 7a). Although distinct individual proteins were identified, their annotation in the same cellular compartment strongly indicates a shared underlying defect across both analyzed cell lines. By overlapping the sets of candidates between the karyotypes, we determined common, polysomy-specific as well as karyotype-dependent p62 interactors in both BafA1 treated and untreated samples. Common hits (18 proteins in BafA1 treated samples, 7 proteins in untreated samples) included p62 interactors that were highly enriched in all cell lines (Supplementary Fig. 7b). Characterization of the common p62 interactors indicated that these are mainly components of the autophagy pathway and also include several mitochondrial proteins (Supplementary Fig. 7c, d, and Supplementary Data 3)[45,46]. Interestingly, although we identified proteins

encoded on the extra chromosomes within the p62 interactome (e.g., MRPL22, chr. 5; PARP4, chr. 13), they were not significantly enriched, consistent with findings from the proxitome analysis (Supplementary Fig. 7a, c, d). Instead, there was a marked over-representation of mitochondrial proteins among the specific interactors identified in polysomic cell lines, but not in the wild type, in both BafA1 treated and untreated samples (Supplementary Fig. 7c, d). The finding was further corroborated by using the robust MitoCarta3.0 inventory[44] of mitochondrial proteins for testing the significance of the enrichment (Supplementary Fig. 7e, f). Since the specific mitochondrial proteins interacting with p62 differed in the two polysomic cell lines, and we detected no significant association with the individual polysomic chromosomes (chr. 5, or 13), we suggest that the observed variations might be attributable to the degree of proteotoxic stress in the analyzed cells. Taken together, these experiments confirm that mitochondrial proteins are the predominant interactors of p62 in cells with additional chromosome(s).

Since the gene encoding p62, *SQSTM1*, is located on chromosome 5, we explored whether its overexpression alone could cause an enrichment of mitochondrial proteins among the interactors. To this end, we transiently overexpressed EGFP-p62 or EGFP alone under the control of a strong CMV promotor[47] in the diploid parental cell line HCT116. This led to approximately 7-fold increase in p62 abundance compared to the control; note that the HCT116 5/4 cell line shows 5-fold increase in p62 abundance (Fig. 4 c, d). We performed p62 co-IP in parental HCT116 with and without p62 overexpression in BafA1 and DMSO-treated conditions, identifying 224-379 proteins, and compared the protein intensities against IgG controls within each condition (Fig. 4e, and Supplementary Fig. 1i, j). Among the interactors, we found NIPSNAP1, NIPSNAP2 (GBAS) and IMMT, which were identified in the pull-downs even without p62 overexpression, suggesting that these mitochondrial proteins interact with p62 independently of cellular karyotype or p62 abundance. We also found HADHB and HSPBP1, which were in previous experiment identified only in the polysomic cell lines. Thus, these proteins may have interacted with p62 due to its increased abundance.

To elaborate the effect of p62 abundance in more detail, we derived robust sets of p62 interactors, which were exclusive in the IP of p62 overexpressing cells and not previously identified as common interactors. This resulted in sets of 17 and 5 interactors for DMSO- and BafA1- treated cells respectively (Supplementary Data 3). We then checked these sets for over-representation of mitochondrial proteins and compared them to those obtained from the p62 pulldowns with polysomic cells. While the p62-overexpressing HCT116 cells showed a mild enrichment of mitochondrial proteins among p62 interactors, the absolute number was rather low (four proteins in DMSO-treated cells, two in BafA1-treated cells) and did not lead to statistically significant differences (Fig. 4f). This is in a contrast with polysomic cells, where all results showed a significant increase in percentage share of mitochondrial hits in both treatment conditions, and higher total numbers of interactors (Fig. 4f). In summary, over-expression of p62 alone cannot explain the significant over-representation of mitochondrial p62 interactors observed in polysomic cell lines.

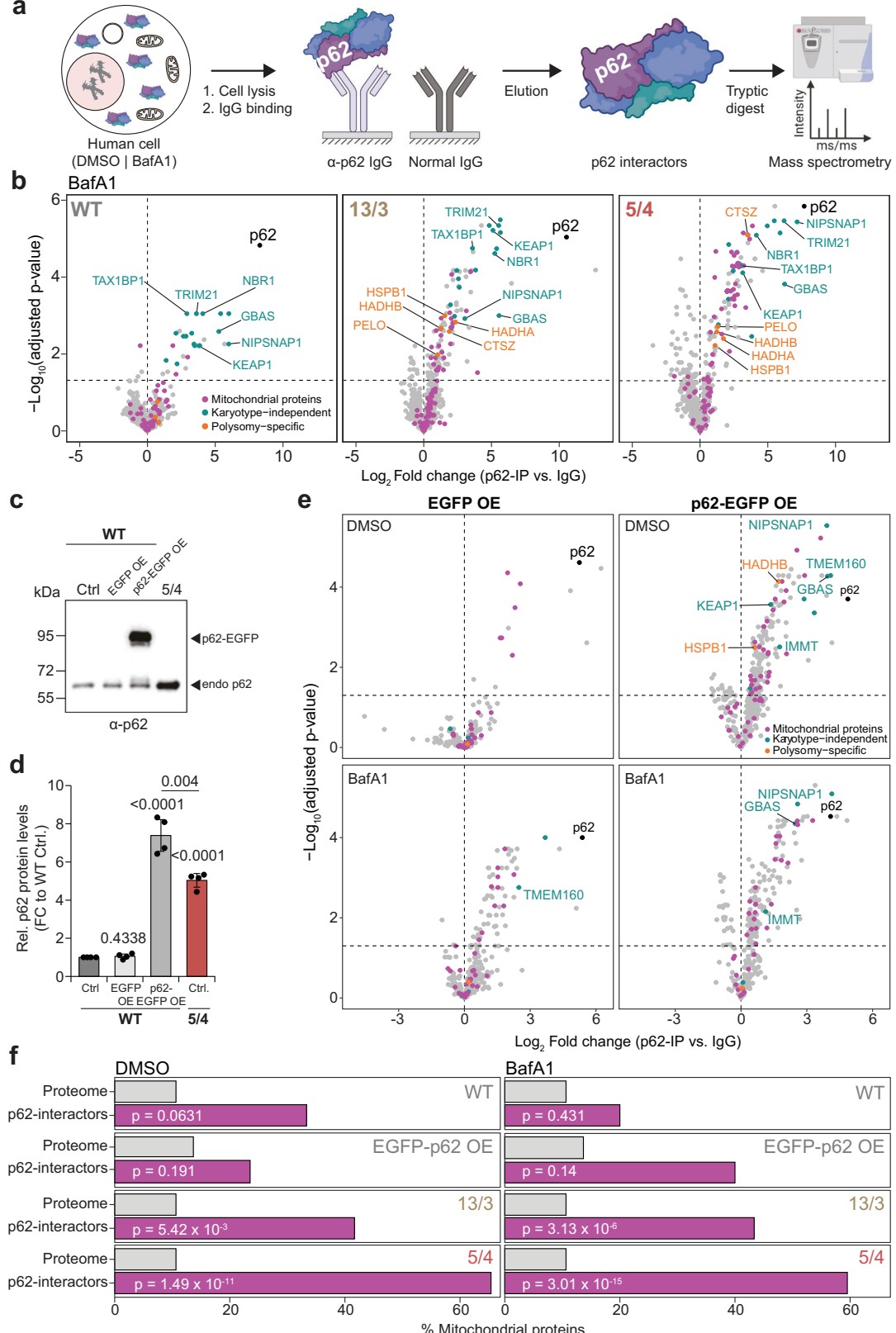

**Chromosome gain impairs mitochondrial architecture and genome**

Since the highly abundant p62 bodies sequester mitochondrial proteins in polysomic cells, we hypothesized that the mitochondria in these cells might be impaired. We reasoned that such an impairment would also lead to the colocalization of p62 with TOMM20 (translocase of outer mitochondrial membrane 20), which is often used as an indicator of mitochondrial damage[48,49]. Analysis of the colocalization of p62 with TOMM20, which was not identified in the IP-MS as an interactor of p62, uncovered low colocalization in HCT116 cells growing in optimal conditions, but a clear increase in polysomic cells (Fig. 5a, b). Quantification of the Pearson correlation coefficient

**Fig. 4 | Co-immunoprecipitation followed by label-free proteomics confirms that mitochondrial proteins are predominant interactors of p62 in polysomic cells. a** Schematic depiction of experimental procedure for p62 pull-down followed by mass spectrometry. Image partly created in BioRender. **b** Volcano plots showing log$_2$-transformed fold change of enriched p62 interactors in WT, 13/3 and 5/4 cells treated with BafA1, highlighting mitochondrial (magenta), karyotype-independent (cyan) and polysomic-specific (orange) interactors. Data is generated from mass spectrometry analysis of $n = 4$ biological replicates (see Supplementary Fig. 1g, h, and Supplementary Data 3). **c** Representative immunoblot and **d** Quantification of p62 expression levels in the diploid parental HCT116 cells (WT-Ctrl.), parental cells transiently overexpressing EGFP (WT-EGFP OE) or p62-EGFP (WT-p62-EGFP OE), and the polysomic 5/4 cells. Data is shown as mean ± s.d. fold change to WT-Ctrl from $n = 4$ independent experiments, and individual replicates are plotted as dots. $P$ values represent two-tailed unpaired Student's $t$-test. **e** Volcano plots showing log$_2$-transformed fold change of enriched p62 interactors in WT-EGFP OE and WT-p62-EGFP OE cells, treated with DMSO or BafA1, highlighting mitochondrial (magenta), karyotype-independent (cyan) and polysomic-specific (orange) interactors. Data is generated from mass spectrometry analysis of $n = 4$ biological replicates (see Supplementary Fig. 1i, j, and Supplementary Data 3). **f** Percentage of mitochondrial proteins in the measured proteome (grey) and the corresponding percentage within the p62 interactome (magenta) of WT, WT-p62-EGFP OE, 13/3 and 5/4 cells. $P$ values represent results of one-sided hypergeometric test, evaluating the differences in representation of mitochondrial proteins relative to the measured proteome.

revealed that the colocalization strongly correlates with the number of surplus protein coding genes (Fig. 5c). The immunofluorescence of TOMM20 also showed that the mitochondrial architecture is altered in polysomic cells compared to the parental cell line, as the fraction of cells with perinuclearly clustered mitochondrial network, which is frequently observed upon stress conditions, such as hypoxia, and is mediated by p62[50], was significantly increased in polysomic cells (Fig. 5d, e). The proportion of cells with perinuclear (i.e., stressed) mitochondria strongly correlated with the number of surplus proteins (Fig. 5f). Due to the accumulation of perinuclear mitochondria, we hypothesized that most of the mitochondria in polysomic cells might be depolarized. To this end, we measured mitochondrial mass and membrane potential by staining the cells with MitoTracker Green FM and Deep Red FM dyes, respectively, and analyzed fluorescence intensity by flow cytometry in the parental diploids, 21/3, 13/3 and 5/4 cells. Surprisingly, the membrane potential, measured as a ratio of MitoTracker DeepRed to MitoTracker Green signal, was slightly higher in polysomic compared to parental cells, while the mitochondrial mass was unchanged (Supplementary Fig. 8a–e). Thus, the gain of a chromosome impairs mitochondria in human cells, but without affecting their membrane potential.

Next, we analyzed the impact of polysomy on the mitochondrial genome. We quantified the mitochondrial DNA copy number (i.e., mtDNA-CN) in polysomic cells in comparison to the parental diploid through quantitative PCR, using primers against the non-coding D-loop and coding sequences of respiratory chain complex or oxidative phosphorylation (OxPHOS) proteins. This analysis revealed that the mtDNA-CN in polysomic cells is reduced to approximately 60 - 80% of the parental cell line (Fig. 5g, and Supplementary Fig. 8f). Given that the mitochondrial mass is unaffected in polysomic cells (Supplementary Fig. 8e), this observation suggests perturbed mitochondrial genome maintenance in response to chromosome gain. Moreover, immunoblotting of selected nuclear-encoded OxPHOS proteins, as well as the inner mitochondrial membrane protein IMMT, which we identified in the p62 IP-MS, revealed more than 20% abundance reduction in polysomic cells (Fig. 5h, i, and Supplementary Fig. 8g). In summary, our data suggest that chromosome gain impairs mitochondrial architecture and genome maintenance and decreases levels of nuclear-encoded respiratory chain components.

## Mitochondrial dysfunction and ROS production are increased in response to chromosome gain

Having established the enrichment of p62 interactors for mitochondrial proteins in polysomic cells, we focused on further characterization of the nature of the mitochondrial defects. Firstly, we asked whether the increased proportion of perinuclearly clustered mitochondria in polysomic cells was indicative of oxidative stress. To this end, we measured the levels of reactive oxygen species (ROS) using normalized CellROX Green intensity. The ROS levels in polysomic cells was significantly higher than in parental cells, and comparable to those of parental cells treated with H$_2$O$_2$ (Supplementary Fig. 8h). Next, we asked whether we could recapitulate the phenotypes exhibited by

polysomic cells by independently inducing ROS or mitochondrial defects in diploid parental cells. Indeed, treatment of the parental HCT116 with H$_2$O$_2$ or with the membrane potential uncoupler CCCP (carbonyl cyanide chlorophenylhydrazone) showed increased p62 foci number and size comparable with the phenotypes observed in polysomic 5/4 cell line (Supplementary Fig. 8i, j). Similarly, the extent of p62 colocalization with mitochondria revealed two-fold increase of the Pearson correlation coefficient in CCCP-treated compared to untreated parental cells, equivalent to levels observed in untreated 5/4 cells (Supplementary Fig. 8k).

Secondly, we evaluated Complex I-dependent oxygen consumption capacity of mitochondria isolated from the diploid parental and 5/4 cells incubated in the presence of cytosolic extracts of reticulocytes with NADH as a substrate[51]. Polysomic cells showed a significant reduction in oxygen consumption to approximately 50% of the parental cells and comparable to CCCP-treated mitochondria from parental cells (Supplementary Fig. 8l, m). Taken together, these observations suggest that mitochondrial homeostasis is significantly altered in response to chromosome gain.

## Cytosolic proteostasis critically affects the formation of p62 bodies and mitochondrial function

Polysomic cells express high levels of p62 and therefore we considered the possibility that the overexpression of p62 alone is sufficient to increase mitochondrial damage in otherwise healthy cells. To this end, we analyzed the impact of p62 overexpression on the formation of p62-bodies and their colocalization with mitochondria in the parental HCT116 cells, using the EGFP and EGFP-p62 constructs (Fig. 6a). While overexpression of EGFP-p62 led to a strong accumulation of p62-positive bodies in the cytosol compared to the control cells overexpressing EGFP only, there was no increase in colocalization with TOMM20 (Fig. 6b-e). Thus, in agreement with the lack of enrichment for mitochondrial proteins interacting with p62 overexpressed in the parental diploids (Fig. 4f), the increased amount of p62 alone is not the cause of the mitochondrial defects in polysomic cells.

Cells with extra chromosomes exhibit impaired cytosolic protein homeostasis partly due to chaperone-mediated protein folding defects[9,52,53]. We hypothesized that this defect caused the observed mitochondrial phenotypes in polysomic cells. To test this hypothesis, we sought to alleviate the protein folding defects by transient overexpression of heat shock protein 27 (HSP27), a small protein chaperone with an antioxidant role in stress resistance; alpha subunit of HSP90 (HSP90α), a molecular chaperone required for the maturation and structural maintenance of many proteins, and the wild type and constitutively active variant of the heat shock transcription factor 1 - HSF1 and ca-HSF1, respectively, the transcriptional factor and master regulator of heat shock response. As a control, we used mock transfected 13/3, 3/3, and 5/4 cells. Overexpression of these constructs was confirmed by Western blot (Supplementary Fig. 9a, d, g). In all instances, we observed decreased numbers of p62-positive foci, but increased foci sizes (Fig. 6f, g, and Supplementary Fig. 9b–i). Pearson correlation coefficient of p62-TOMM20 colocalization revealed

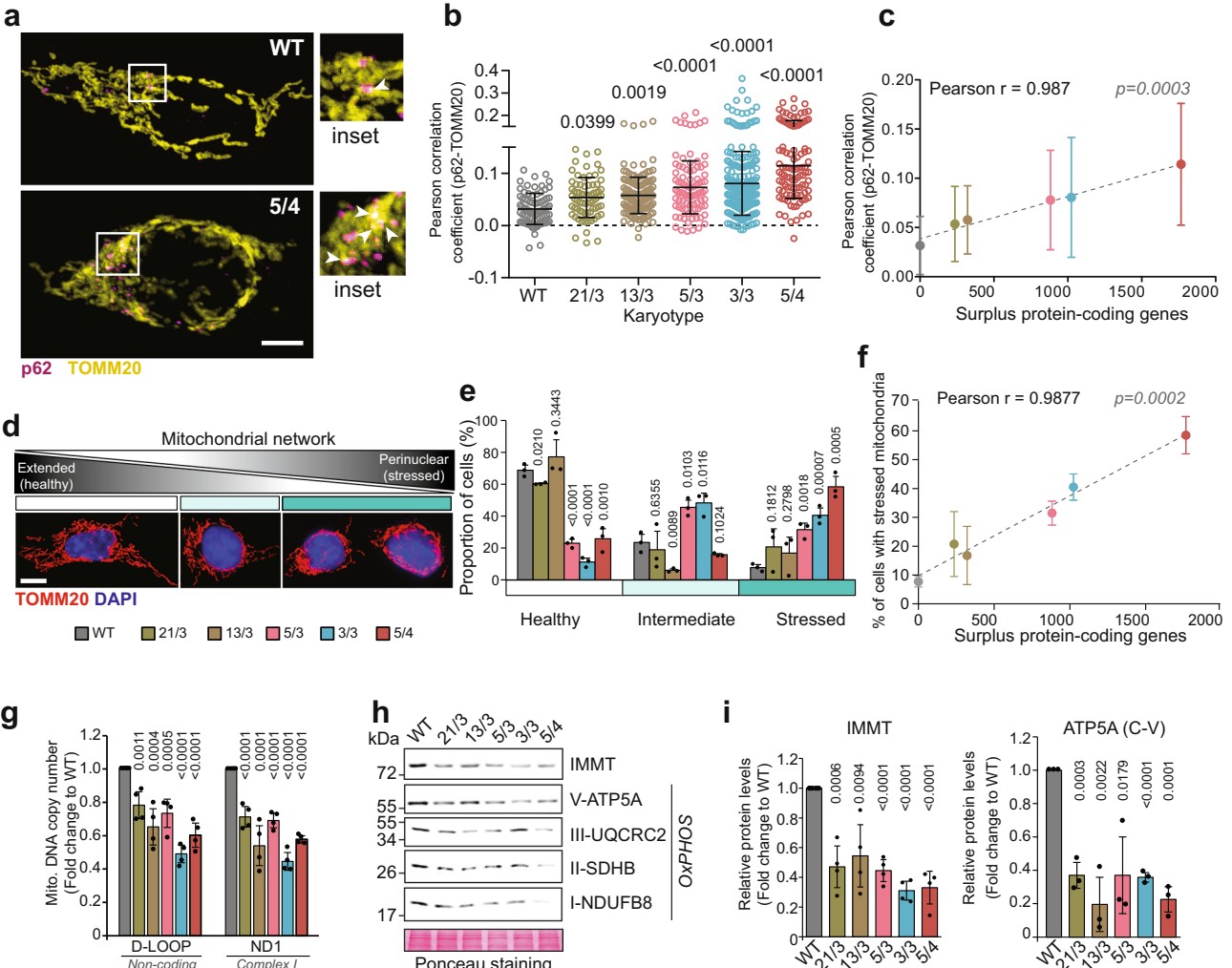

**Fig. 5 | Polysomic cells exhibit increased mitochondrial defects.**
**a** Representative confocal images of p62 (magenta) and mitochondrial outer membrane translocase TOMM20 (yellow) colocalization in diploid parental (WT) and a polysomic (5/4) cell line. Magnified inserts are shown. White arrow heads indicate p62-TOMM20 colocalization. Images are collapsed from Z-stacks. Scale bar is 10 μm.
**b** Quantification of the Pearson correlation coefficient (PCC) for p62-TOMM20 colocalization in WT, 21/3, 13/3, 5/3, 3/3 and 5/4 cells. Values of individual cells from $n = 3$ independent experiments and mean ± s.d. are shown. *P*-values represent one-way ANOVA followed by Sidak's multiple comparisons test. **c** Correlation of the mean ± s.d. PCC for p62-TOMM20 colocalization from (**b**) with the number of surplus protein-coding genes in all the cell lines. Two-tailed *P*-value is indicated.
**d** Representative confocal images of the different mitochondrial architecture (red) observed in the cells. DAPI (blue) is used as a nuclear stain. Images are collapsed from Z-stacks. Scale bar is 10 μm. **e** Quantification of the cell fractions with each mito-chondrial form, classified as healthy (extended network), intermediate (minimally

extended network) and stressed (perinuclearly clustered network) as shown in (**d**). Data represent mean ± s.d. percentage of $n = 3$ independent experiments, and individual replicates are plotted as dots. *P* values represent two-tailed unpaired Student's *t*-test. **f** Correlation of the mean ± s.d. proportion of cells harboring stressed (peri-nuclear) mitochondria from (**e**) with the number of surplus protein-coding genes in all the cell lines. Two-tailed *P*-values are indicated. **g** Relative mitochondrial DNA (mt-DNA) copy number determined by qPCR of the non-coding mt-DNA D-Loop and respiratory chain complex 1 subunit ND1, normalized to the nuclear ß₂-microglobulin gene. Data is shown as mean ± s.d. fold change to WT from $n = 5$ independent experiments, and individual replicates are shown as dots. *P* values represent two-tailed unpaired Student's *t*-test. **h** Representative immunoblot and **i** Quantification of expression levels of selected mitochondrial proteins. Ponceau staining is used as loading control. Data is shown as mean ± s.d. fold change to WT from $n = 3–4$ independent experiments, and individual replicates are plotted as dots. *P* values represent two-tailed unpaired Student's *t*-test.

significantly reduced colocalization in the 13/3 and 3/3 cells over-expressing the constructs, while the 5/4 cells showed no rescue (Fig. 6h, and Supplementary Fig. 9c, f, i). We conclude that the pro-teotoxic stress and mitochondrial defect can be rescued by transient overexpression of cytosolic chaperones, but only in polysomic cells suffering from mild proteotoxic stress. Severe proteotoxic stress, as in the 5/4 cell line with high numbers of extra protein coding genes, could not be fully rescued. The increased size of the p62 foci observed after overexpression of the heat shock factors could also reflect exacer-bated protein aggregation. However, staining the cells with the aggresome dye showed that the overexpression of heat shock proteins significantly decreased aggresome formation in all tested aneuploid cell lines (Fig. 6i–k). Our data suggest that impaired protein folding

and chronic overexpression of extra proteins in polysomic cells facil-itate formation of cytosolic p62-positive bodies that sequester mito-chondrial proteins, thus impairing mitochondrial functions. Overexpression of cytosolic chaperones alleviates the formation of toxic protein aggregates by promoting their consolidation into fewer larger, less mobile structures that are less harmful. This, in turn, miti-gates the associated defects in polysomic cells[54,55].

## Mitochondrial protein import is diminished in cells with extra chromosomes

Our data suggests that enrichment of mitochondrial proteins among the p62 interactors in polysomic cells arises due to the accumulation of unimported mitochondrial precursor proteins in cytosolic aggregates.

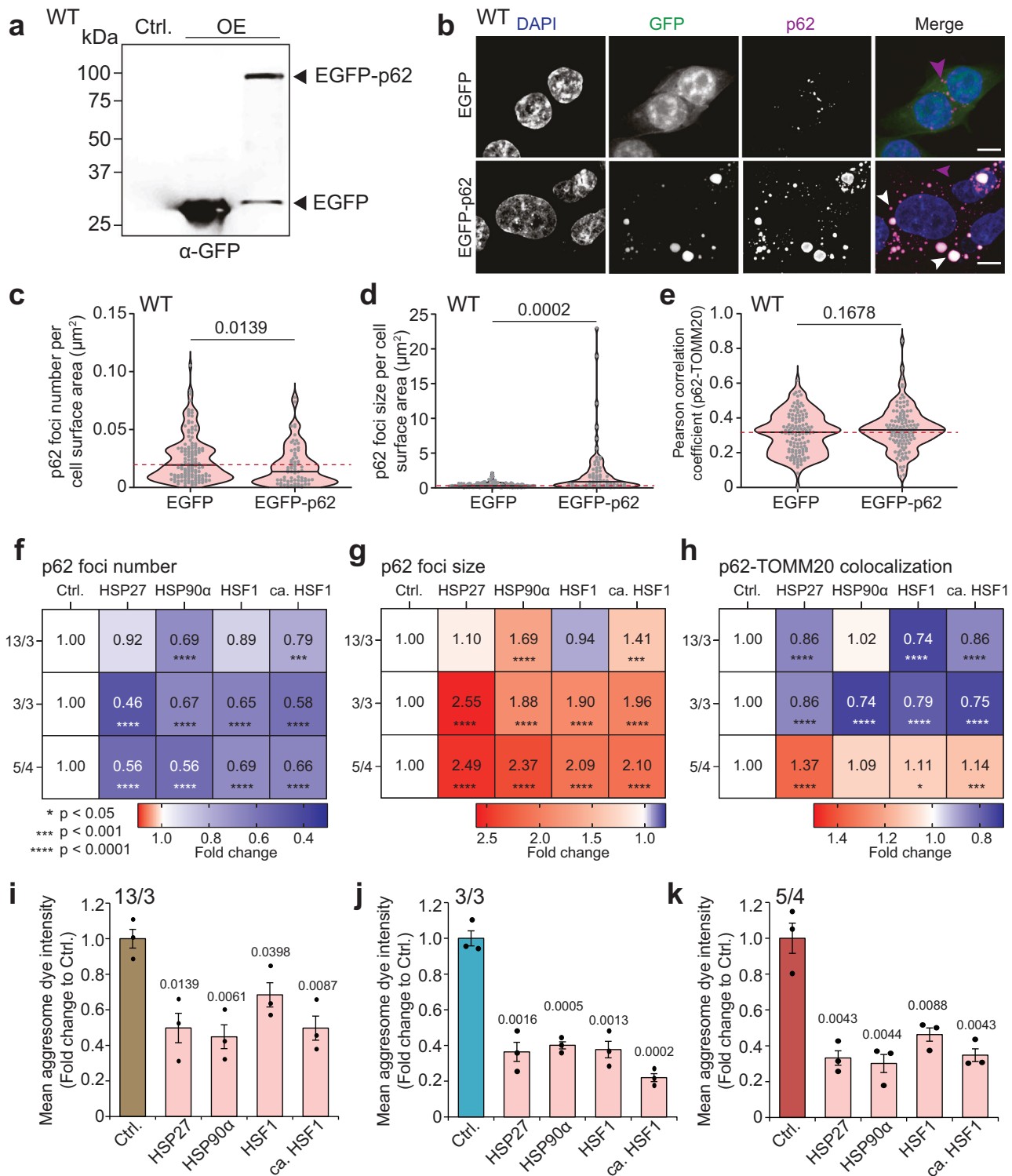

More than 99% of mitochondrial proteins are synthesized by cytosolic ribosomes as precursors and guarded by cytosolic chaperones in their unfolded state to facilitate their import into mitochondria, where they are processed into mature forms (Fig. 7a[56]). We hypothesized that the limited chaperone activity in polysomic cells[9,35] could lead to the accumulation of mitochondrial precursor proteins in the cytosol, making them prone to misfolding, aggregation and incorporation into protein aggregates. These aggregates, in turn, may impair mitochondrial biogenesis and function[57–60]. To test the above scenario, we analyzed the expression of three selected mitochondrial proteins that were identified among p62 interactors through the IP-MS: HADHA,

HADHB, and MRPL45, by Western blot in whole-cell lysates of the five different polysomic cell lines and parental diploid cells. Strikingly, we observed an increased accumulation of the precursor forms of each of the proteins in whole-cell lysates of the polysomic cells (Fig. 7b). This indicates that mitochondria of polysomic cells exhibit import defects, despite their normal membrane potential (Supplementary Fig. 8a-d). To validate this observation by an orthogonal approach, we isolated mitochondria from the cells and performed kinetic import of radiolabeled mitochondrial substrates su9-DHFR and human SOD2 (Fig. 7c, d, and Supplementary Fig. 10[61]). The mitochondria enrichment was comparable in all cell lines, as confirmed by Western blot

**Fig. 6 | Limited chaperone capacity underlies p62 deposits accumulation and mitochondrial defects, which are ameliorated by transient overexpression of protein folding factors. a** Representative immunoblot of GFP showing EGFP and EGFP-p62 overexpression in diploid parental (WT) cells. **b** Representative confocal images of p62 (magenta) and EGFP (green) foci visualization in WT cells transiently transfected with EGFP and EGFP-p62 plasmids. DAPI (blue) is used as a nuclear stain. Images are collapsed from Z-stacks. Scale bar is 10 μm. Quantifications of (**c**) p62 foci number per cell surface area in $\mu m^2$, **d** p62 foci size per cell surface area in $\mu m^2$, and **e** Pearson correlation coefficient for p62-TOMM20 colocalization in respective samples. Violin plots indicate mean and distribution of values, while dots represent individual cells from $n = 3$ independent experiments. *P*-values represent two-tailed unpaired *t* test with Welch's correction. Heatmaps showing (**f**) p62 foci number per cell surface area in $\mu m^2$, **g** p62 foci size per cell surface area in $\mu m^2$, and **h** Pearson

correlation coefficient for p62-TOMM20 colocalization in 13/3, 3/3 and 5/4 polysomic cell lines transiently transfected with HSP27, HSP90α-Flag, EGFP-HSF1 and constitutively active HSF1 (see Supplementary Fig. 9). Values represent fold changes to mock transfected cells (Ctrl.). *P* values represent non-parametric ANOVA followed by Dunn's multiple comparisons test (**f** and **g**) and one-way ANOVA followed by Sidak's multiple comparisons test (**h**). * $p < 0.05$, *** $p < 0.001$, **** $p < 0.0001$ (for exact *p*-values, see Supplementary Fig. 9c, f, i). **i−k** Quantification of the aggresome dye fluorescence intensity of 13/3, 3/3, and 5/4 cells transiently transfected with HSP27, HSP90α-Flag, EGFP-HSF1 and constitutively active HSF1. Data is shown as mean ± s.e.m fold change to mock transfected cells (Ctrl.) from $n = 3$ independent experiments, and individual replicates are plotted as dots. *P* values represent two-tailed unpaired Student's *t*-test.

(Supplementary Fig. 10b). Kinetic import analysis revealed a delayed import of both su9-DHFR and human SOD2 into isolated mitochondria from polysomic cells (Fig. 7c, d, and Supplementary Fig. 10c, d). We propose that cytosolic proteotoxic stress due to defective protein folding is associated with mild, but chronic reduction of mitochondrial protein import and impaired mitochondrial function in aneuploid cells. The unimported mitochondrial precursor proteins accumulate in the cytosol, where they get sequestered into p62-positive bodies (Fig. 7e).

## Discussion

Protein homeostasis is crucial for cellular and organismal functions. Numerical chromosomal aneuploidy, which is associated with various pathologies, alters the abundance of the proteins encoded on affected chromosomes. The low, but chronic overexpression of hundreds of unneeded proteins in cells with extra chromosomes alters proteome landscape of the cells and triggers aneuploidy-associated stress response. As a consequence, human cells with additional chromosomes suffer from proteotoxic stress, as documented by increased sensitivity to chaperone inhibitors, accumulation of ubiquitin- and p62-positive cytosolic bodies and increased lysosomal stress[8,9,14–16,18]. By analyzing the phenotypes of five isogenic cell lines with different extra chromosomes, we show that the number and size of p62-positive cytosolic bodies increases compared to the parental diploids, and this increase tightly correlates with the number of extra protein-coding genes in polysomic cells (Fig. 1a−g). Similarly, the degree of aneuploidy quantified as aneuploidy score in cancer datasets positively correlates with p62 abundance and p62 essentiality (Fig. 1h−j). Thus, p62 expression scales with the protein imbalance caused by chromosome number changes. Acute aneuploidy after chromosome missegregation may lead to p62 accumulation through transient lysosomal stress and autophagy overburdening[14,15]. In contrast, chronic aneuploidy does not impair autophagy flux and triggers increased transcription of the *SQSTM1* gene in both model cell lines and in cancer datasets ([8,19], Fig. 1). The multifaceted protein p62 has a complex role in maintenance of protein homeostasis, and in response to various stresses[62]. Understanding what triggers accumulation of p62 in aneuploidy will be crucial to elucidate its link to cancer.

We hypothesized that analysis of the protein content of the cytosolic p62 bodies in aneuploid cells will provide an insight into the changes caused by the proteotoxic stress in these cells. Proteomic analysis of p62 proxitome and interactome revealed a marked complexity of their composition. Besides previously known interactors, we found and validated several new, aneuploidy-specific interactors, and a strong enrichment for mitochondrial proteins (Figs. 2−4, and Supplementary Fig. 3−7). Remarkably, we did not observe any significant enrichment for proteins encoded on the extra chromosomes, suggesting that p62 does not specifically sequester these supernumerary proteins. Another aspect is the lack of an enrichment of mitochondrial proteins in the spatially restricted p62-proxitome of autophagosomes. This suggests that p62-associated mitochondrial proteins in polysomic

cells are not recycled via autophagy, but rather remain in the cytosol or are removed by different means, e.g., by extrusion from the cells. In future, profiling of autophagosomes via proximity labeling approaches with other autophagy cargo carriers[42] as well as comprehensive analyses of the fate of unimported mitochondrial precursor proteins and damaged mitochondria in polysomic cells will provide further insights about changes in mitochondrial homeostasis in response to aneuploidy.

The p62 bodies often localized near mitochondrial surface in polysomic cells (Fig. 5a−c), a feature which correlates with mitochondrial damage, but not necessarily with mitophagy[63]. Indeed, cells with extra chromosomes contain stressed, less functional mitochondria, and impaired mitochondrial metabolism, reduced levels of nuclear-encoded mitochondrial proteins and mtDNA (Fig. 5, and Supplementary Fig. 8). Aneuploidy has previously been linked to metabolic stress due to increased ROS levels and altered mitochondrial metabolism[64,65]. Additionally, aneuploid cells exhibit a metabolic shift from oxidative phosphorylation to glycolysis[66], though the underlying causes remain unclear. This shift parallels the Warburg effect observed in many cancers. In Drosophila, saturated autophagy in aneuploid cells causes accumulation of dysfunctional mitochondria, increased ROS production and subsequent cellular senescence[67]. Mitochondrial dysfunction is also frequent in Down syndrome, which is caused by trisomy of chromosome 21. Here, downregulation of nuclear-encoded mitochondrial genes, reduced efficiency of energy production, reduced oxygen consumption, and altered mitochondrial morphology were observed[68]; these features we also observed in engineered polysomic cells regardless of the identity of the extra chromosome. While increased expression of genes encoded on the aneuploid chromosomes, such as the DYRKA1 kinase located on chromosome 21[69], likely contributes to mitochondrial phenotypes, our results suggest that the cytosolic proteotoxic stress caused by production of extra proteins may be another factor causing mitochondrial dysfunction independently of specific karyotype.

What is causing the mitochondrial defects in polysomic cells? Mitochondria contain their own DNA, but 99% of mitochondrial proteins are encoded in the nucleus, synthesized in the cytosol, and imported into mitochondria by dedicated transport pathways[56]. Precursor proteins are often metastable and aggregation-prone and must be kept in their unfolded state by chaperones for import. This may be a challenge in cells with extra chromosomes that produce hundreds of superfluous proteins, which also require chaperones for their folding. Indeed, we observed increased accumulation of mitochondrial precursor proteins in the cytosol, and protein import into mitochondria was delayed in polysomic cells (Fig. 7a−d). Mitochondrial inner membrane-localized proteins and mitoribosomal proteins, which are usually hydrophobic in nature, pose severe challenges to cytosolic proteostasis when their import into mitochondria is impaired. Their sequestration into aggregates acts as a protective measure to minimize their negative effect[70]. We found that these proteins were particularly enriched in the p62-bodies in aneuploid cells (Figs. 2, 4, and

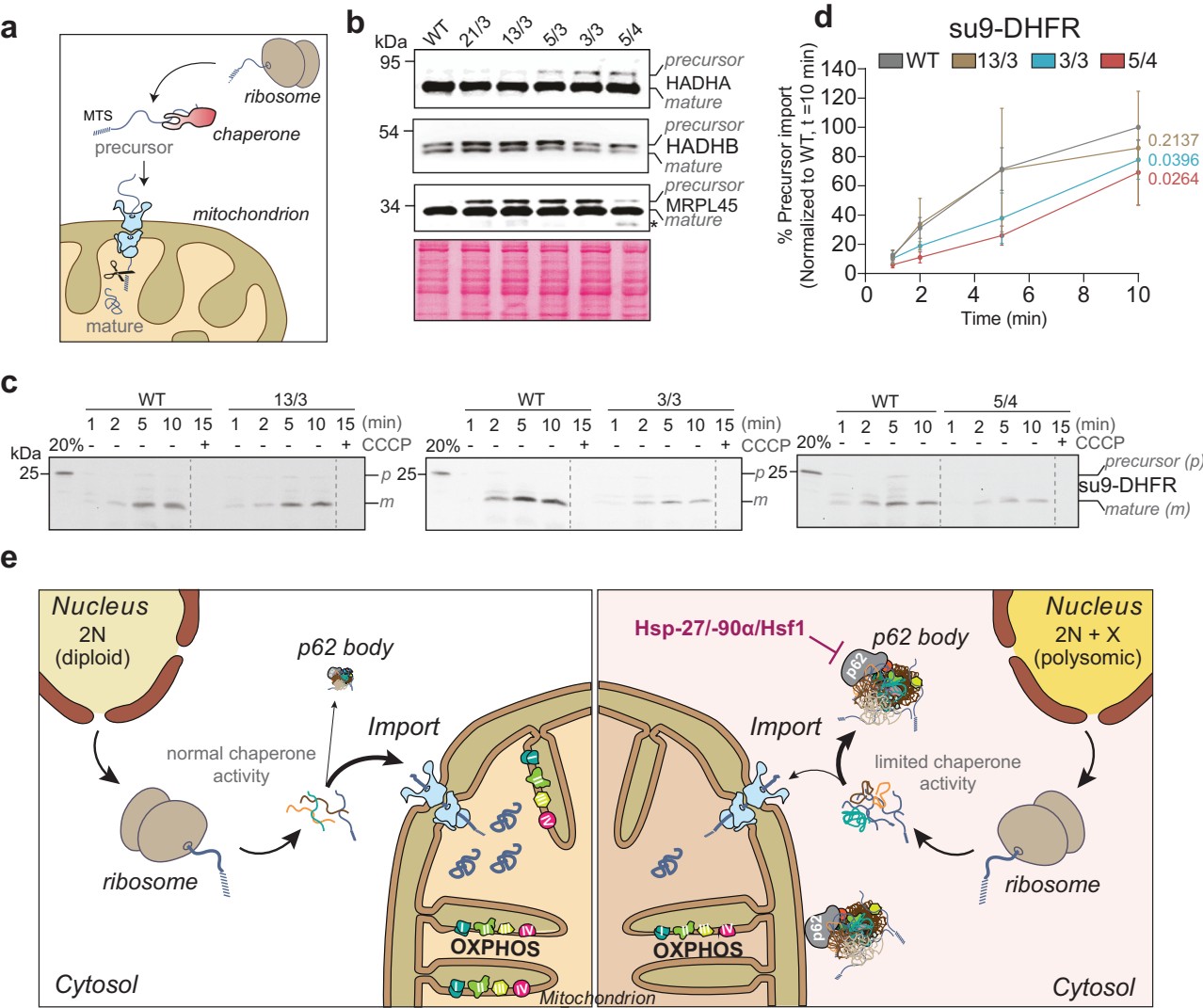

**Fig. 7 | Chromosome gain impairs mitochondrial precursor protein import.**
**a** Schematic depiction of the process of posttranslational protein import into mitochondria. Most mitochondrial proteins are synthesized as precursors by cytosolic ribosomes and guarded by chaperones in their unfolded state for import into mitochondria, where the pre-sequences are cleaved, and the proteins folded into their native state. **b** Immunoblot of selected mitochondrial proteins (from Fig. 4b) showing precursor and mature forms. Ponceau staining is used as a loading control. *Degradation product. **c** Representative images showing import kinetic of synthesized $^{35}$S-methonine-labeled su9-DHFR into mitochondria isolated from WT, 13/3, 3/3 and 5/4 cells. The su9-DHFR precursor is processed upon reaching the matrix (i.e., mature form). CCCP, which depletes mitochondrial membrane potential thereby preventing import, is used as a negative control. 20% of the synthesized substrate (precursor) is loaded for comparison. All samples are resolved by SDS-PAGE and visualized by autoradiography. **d** Quantification of the import kinetic from (**c**). Data represents mean ± s.e.m. of $n = 4$ (for 13/3, 3/3 and 5/4

cell lines) and $n = 12$ (for WT cell line; four for each polysomic cell line) independent import assays. Each polysomic cell line is run alongside the parental cell line per experiment as shown in (**c**). The data is normalized to WT, t = 10 mins (set to 100%). P-values represent one-tailed paired t-test. **e** Proposed model of aneuploidy-induced disruptions of cellular proteostasis and their effect on mitochondria. The gain of extra chromosome leads to translation of excess proteins, which places a severe burden on cytosolic proteostasis. Aberrant cytosolic proteostasis due to the expression of surplus proteins limits chaperone-mediated activity, thereby impairing mitochondrial protein import and leading to the accumulation of mitochondrial precursor proteins in the cytosol, where they are sequestered into p62-positive bodies (right). The p62-positive bodies do not only reside in the cytosol, but also associate with the mitochondrial outer membrane and interfere with mitochondrial functions. Overexpression of protein folding factors decreases the formation of p62-positive bodies and their association with mitochondria.

Supplementary Figs. 3, 7). We hypothesize that the impaired protein folding and imbalanced protein homeostasis due to aneuploidy induces protein aggregation[8,14,18,35,52,53] in cytosolic p62-positive bodies, which in turn sequester unimported mitochondrial proteins. The reduced import of essential mitochondrial proteins into mitochondria impairs mitochondrial health (Fig. 7e). It remains unclear what exactly triggers the formation of p62-positive cytosolic bodies enriched with mitochondrial proteins. One hypothesis is that impaired cytosolic proteostasis in polysomic cells may disrupt mitochondrial precursor delivery to mitochondria, which leads to formation of p62 bodies. Alternatively, proteostasis impairment may directly promote the

formation of the p62 bodies, which have a high propensity to interact with metastable mitochondrial precursor proteins. Additionally, factors such as oxidative stress and alterations in lipid metabolism could play a role. The precise molecular mechanisms underlying these phenomena are currently under investigation.

Recent years have brought progress in our understanding of the intimate link between mitochondrial function and cytosolic protein homeostasis[71,72]. Cytosolic protein folding and quality control mechanisms ensure that nuclear-encoded proteins are correctly targeted to their mitochondrial destinations and any disruption of this process exacerbates their misfolding and aggregation[60,71]. Disruptions

in cytosolic proteostasis can affect mitochondrial morphology and distribution due to disruption of mitochondrial dynamics. An increased oxidative stress due to aberrant mitochondria in aneuploid cells was recently found to contribute to additional genomic instability[73,74]. We propose that the mitochondrial dysfunction described here and characterized by increased ROS production contributes to this phenomenon. Taken together, aneuploidy, an omnipresent feature of cancer cells, impairs protein import into mitochondria and triggers sequestration of mitochondrial proteins into cytosolic aggregates, thereby promoting mitochondrial dysfunction. Our work sheds new insight into consequences of aneuploidy and provide novel, physiologically relevant model system to study the link between cytosolic protein imbalance and mitochondrial functions.

## Methods

### Cell lines and culture conditions
Polysomic cell lines were derived from the near-diploid human colon carcinoma cell line HCT116 (ATCC No. CCL-247) by microcell-mediated chromosome transfer as previously described[8,34]. Parental HCT116 (WT), HCT116 21/3 (trisomy of chromosome 21), 13/3 (trisomy of chromosome 13), 5/3 (trisomy of chromosome 5, expressing H2B-GFP), 3/3 (trisomy of chromosome 3, expressing H2B-GFP), and 5/4 (tetrasomy of chromosome 5) were cultured in Dulbecco's Modified Eagle Medium (DMEM; Gibco) supplemented with 10% Fetal Bovine Serum and 5% Penicillin/Streptomycin and incubated at 37 °C with 5% $CO_2$ atmosphere. All cell lines were used up to a maximum of five passages before replacement with fresh ones and tested regularly for mycoplasma contamination using MycoStrip tests (InvivoGen). Bafilomycin A1 was applied at 100 nM for 6 h, CCCP at 10 μM for 6 h.

### Transient cell line transfection
Transient expression of the following plasmids HSP90α, HSP27, HSF1, ca. HSF1, pEGFP-N3, EGFP-p62 was performed using Lipofectamine 2000 (Thermo Fisher Scientific) according to the manufacturer's instructions. Transfection medium was replaced 6 h post-transfection and transfected cells were analyzed 48 h post-transfection. A list of plasmids used is provided in Supplementary Table 1.

### Stable cell line transfection
HCT116 WT, 13/3 and 5/4 cell lines stably expressing myc-APEX2-p62 were generated by lentiviral transduction. Briefly, HEK 293 T cells were transiently transfected with the packaging plasmids pHDM-Hgpm2, pHDM-Tat1b, pHDM-VSV-G, and pRC-CMV-Rev1b[75] and the expression plasmids using Lipofectamine 2000 (Thermo Fisher Scientific) according to the manufacturer's instructions. Fourty-eight hours post-transfection recipient cells were infected with the viral supernatant in the presence of 8 μg/ml polybrene according to the manufacturer's instructions (Sigma-Aldrich). myc-APEX2-p62 transduced cells were selected in cell culture media supplemented with 2 μg/ml puromycin (Thermo Fisher Scientific) A list of plasmids used is provided in Supplementary Table 1.

### Analysis of p62 interactome by co-immunoprecipitation-based quantitative proteomics
For immunoprecipitation, cell lysates were prepared by lysis in buffer [150 mM NaCl, 50 mM Tris-HCl (pH 7.5), 5% Glycerol, 1% IGPAL-CA-630, 1 mM MgCl₂] supplemented with 1 mM PMSF and 1X Protease inhibitor cocktail (Merck). Samples were incubated on an overhead rotator for 20 min and centrifuged at 500 g for 10 min at 4 °C to remove cell debris. Total protein concentration was determined in the supernatant fraction by Bradford assay. SQSTM1/p62 antibody (Santa Cruz, sc-28359) and normal mouse IgG (Santa Cruz, sc-2025) were added separately to 1 mg of total protein at 2 μg and 0.25 μg per reaction, respectively, and incubated at 4 °C on rotation overnight. Subsequently, Dynabeads™ Protein G magnetic beads (Thermo Fisher

Scientific) were added to the lysate/antibody mixture and incubated at RT for 30 min on rotation. Immunoprecipitated proteins were washed 3X with wash buffer [150 mM NaCl, 50 mM Tris-HCl (pH 7.5), 5% Glycerol] and eluted by an on-bead tryptic digest using elution buffer [2 M Urea (prepared with 50 mM NH₄HCO₃ solution), 50 mM Tris-HCl (pH 7.5), 1 mM DTT, 5 μg/ml Trypsin] at 37 °C for 1 h with shaking at 600 rpm. Samples were then alkylated overnight with 5 mM iodoacetamide (IAA) at 37 °C and shaking at 600 rpm. Digested and alkylated peptides were acidified with 1% TFA, desalted and purified on C18 stage tips for analysis by mass spectrometry.

### Analysis of p62 proxitome by proximity biotinylation-based quantitative proteomics
APEX2-mediated biotinylation of the p62 proxitome in cells was performed by modifying the method described by ref. 42. Briefly, cells were supplemented with 500 mM biotin-phenol (Iris Biotech) for 30 min at 37 °C followed by addition of 1 mM $H_2O_2$ for 1 min at RT. Biotinylation reaction was stopped by washing the cells three times with quencher solution [1 mM sodium azide, 10 mM sodium ascorbate and 5 mM Trolox in DPBS] followed by three washes with PBS. Cells treated with $H_2O_2$ only were included as controls for the biotinylation reaction. Cells were then harvested by scrapping and lysed by dounce-homogenization and sonication in RIPA buffer containing quenching components [50 mMTris, 150 mM NaCl, 0.1% SDS, 0.5% sodium deoxycholate, 1% Triton X-100, 1x protease inhibitors (Merck), 1x PhosStop (Merck), 1 mM sodium azide, 10 mM sodium ascorbate and 1 mM Trolox]. Cleared lysates were obtained by centrifugation at 10000 g for 10 min at 4 °C and incubated overnight on pre-equilibrated Streptavidin-Agarose beads (Sigma). The next day, samples were washed three times with RIPA buffer containing quenching components, followed by three washes in 3 M Urea buffer (prepared with 50 mM NH₄HCO₃ solution). Samples were then incubated with 5 mM TCEP for 30 min at 55 °C and alkylated with 10 mM IAA for 20 min at RT. Alkylation was quenched by the addition of 20 mM DTT and samples were washed twice with 2 M Urea buffer (prepared with 50 mM NH₄HCO₃ solution) and digested overnight at 37 °C using 1 μg of trypsin per 20 μl beads. Digested peptides from the supernatants were collected and pooled together with supernatants from two times washes of the beads using 2 M Urea buffer. The pooled samples were then acidified with 1% TFA and concentrated by vacuum centrifugation. Finally, the digested peptides were desalted and purified on C18 stage tips for analysis by mass spectrometry.

### Analysis of p62 cargo candidates engulfed in autophagosomes by proximity biotinylation-based quantitative proteomics
Autophagosomal content profiling for p62 cargo candidates was performed as described by[42]. Briefly, APEX2-mediated biotin labeled cells were incubated on an overhead rotator for 20 min at 4 °C in homogenization buffer I [10 mM KCl, 1.5 mM MgCl₂, 10 mM HEPES-KOH and 1 mM DTT pH 7.5]. Cells were then lysed by dounce homogenization and mixed at a ratio of 1:5 with homogenization buffer II [375 mM KCl, 22.5 mM MgCl₂, 220 mM HEPES-KOH and 0.5 mM DTT pH 7.5]. Cleared lysates obtained by centrifugation at 600 g for 10 min were treated with 100 mg/ml Proteinase K (PK) and 1 mM CaCl₂ for 1 h at 37 °C, before inhibition of PK activity by adding 10 mM PMSF to the samples. To account for Proteinase K resistant proteins, samples additionally treated with 0.1% RAPIGest (RP) were added as control. Samples were centrifuged at 17000 g for 15 min at 4 °C. The supernatant was removed, and pellets were resuspended in RIPA buffer containing quenching components, followed by brief sonication and centrifugation at 10000 g for 10 min h at 4 °C. Supernatants were then incubated on pre-equilibrated Streptavidin-Agarose (Sigma) overnight at 4 °C. Samples were then washed three times in RIPA buffer containing quenching components and three times in 3 M Urea buffer (prepared with 50 mM NH₄HCO₃ solution). Subsequently, samples were

incubated with 5 mM TCEP for 30 min at 55 °C and alkylated with 10 mM IAA for 20 min at RT. Alkylation was quenched by the addition of 20 mM DTT and samples were washed twice with 2 M Urea buffer (prepared with 50 mM $NH_4HCO_3$ solution) and digested overnight at 37 °C using 1 μg of trypsin per 20 μl beads. Digested peptides from the supernatants were collected and pooled together with supernatants from two times washes of the beads using 2 M Urea buffer. The pooled samples were then acidified with 1% TFA and concentrated by vacuum centrifugation. Finally, the digested peptides were desalted and purified on C18 stage tips for analysis by mass spectrometry.

To validate the PK protection assay by Western blot, the cleared lysates obtained by centrifugation at 600 g for 10 min were treated with 30 mg/ml PK and 1 mM $CaCl_2$ and incubated for 30 min at 37 °C. Control reactions consisting of no digestion, digestions with PK and 0.1% RP or treatment with 0.1% RP only were included. Samples were then boiled in Laemmli buffer and subjected to SDS-PAGE.

### Analysis of total proteome by tandem mass tag (TMT)-based quantitative proteomics

Preparation of cell lysates and labelling of peptides with TMT isobaric mass tags was carried out using the commercially available kit and reagents (Thermo Fischer Scientific, #90110) according to the manufacturer's protocol. Briefly, cell lysates were obtained by suspension of cells in 10% SDS buffer (prepared with 100 mM triethyl ammonium bicarbonate) followed by sonication. Samples were centrifuged at 16,000 g for 10 min at 4 °C and protein concentration determined from the supernatant with a BCA assay (Thermo Fisher Scientific). Approximately 50 μg of protein was reduced with 200 mM TCEP for 1 h at 55 °C and alkylated with 10 mM IAA for 30 min in the dark at RT. Proteins were then acetone precipitated overnight and digested with 1.5 μg proteomics-grade trypsin (Sigma-Aldrich) at 37 °C overnight. Subsequently, 25 μg of digested peptides were labelled with TMT10plex isobaric tags (Thermo Fisher Scientific) for 1 h at RT followed by quenching with 5% hydroxylamine for 15 min at RT. TMT labeled samples were adjusted to equal amounts through a LC-MS test run (with 2 μl of each sample pooled together, desalted, and purified on C18 membranes), before fractionation of the total sample. Individually labelled samples were then pooled, fractionated into 8 fractions using the High pH Reversed-Phase Peptide Fractionation Kit (Thermo Fisher Scientific) according to the manufacturer's instructions protocol and dried by vacuum centrifugation.

### Liquid chromatography-tandem mass spectrometry

Peptides were resuspended in 9:1 mixture of buffer A [0.1% formic acid] and buffer A* [2% acetonitrile, 0.1% trifluoro acetic acid]. Half of the sample was separated on 50 cm columns packed in house with ReproSil-Pur C18-AQ 1.9 μm resin (Dr. Maisch GmbH). Using an EASY-nLC 1200 ultra-high-pressure system connected in-line to a Q-Exactive HF Mass Spectrometer (Thermo Fisher Scientific), liquid chromatography was conducted. Peptides were introduced in buffer A [0.1% formic acid] and eluted with a non-linear gradient of 5–60% buffer B [0.1% formic acid, 80% acetonitrile] at a rate of 300 nl/min over 90 min. Column temperature was maintained at 60 °C. Data acquisition alternated between a full scan (60 K resolution, maximum injection time of 20 ms, AGC target of 3e6) and 15 data-dependent MS/MS scans (15 K resolution, maximum injection time of 80 ms, AGC target of 1.6e3). The isolation window was 1.4 m/z, and normalized collision energy was 28. Dynamic exclusion was set to 30 sec.

TMT-labelled peptides were separated as described above except that a 180 min gradient was used for elution. MS data was acquired in data-dependent mode using the following settings. Full MS scans (120 K resolution, maximum injection time of 80 ms, AGC target of 3e6) were alternated with a set of maximally 15 data-dependent MS/MS scans (60 K resolution, maximum injection time of 100 ms, AGC target of 2.0e3). The isolation window was 0.7 m/z, and normalized collision energy was 32. Fixed first mass was set to 100 m/z. Dynamic exclusion was set to 30 sec.

### Mass spectrometry data analysis

Raw mass spectrometry data was processed using using MaxQuant (2.0.1.0). Peak lists were searched against a Uniprot database (UP000005640_9606), alongside 262 common contaminants using the Andromeda search engine. A 1% false discovery rate was applied for both peptides (minimum length: 7 amino acids) and proteins. The "Match between runs" (MBR) function was activated, with a maximum matching time window of 0.7 min and an alignment time window of 20 min. Relative protein quantities were calculated using the MaxLFQ algorithm, requiring a minimum ratio count of two. The calculation of iBAQ intensities was enabled.

### Computational analysis of mass spectrometry data

The protein groups identified in each mass spectrometry data set were processed and analyzed in parallel using the R programming language (version 4.3.3; R Core Team 2024). First, the MaxQuant output was filtered to remove contaminants, reverse hits, proteins identified by site only as well as proteins that were identified in less than N-1 replicates of every condition, where N is the number of replicates (N ≥ 3). The label-free quantification (LFQ) or corrected reporter ion intensities of the robustly identified proteins were log2-transformed subsequently. For the global proteome data set only, the protein intensities were normalized between samples using fast cyclic loess normalization as implemented in the R package limma[76] (version 3.58.1). Lastly, missing values were imputed by sampling N values from a normal distribution (seed = 12345) and using them as replacements only when there are no valid values in a group of replicates of a condition or if there is only one valid value which is below the first tertile of intensities in the data set. Different for each data set, the mean of this normal distribution corresponds to the 1% percentile of LFQ intensities, and its standard deviation is determined as the median of protein intensity sample standard deviations calculated within and then averaged over each group of replicates.

Principal component analysis was carried out for each data set using the package pcaMethods[77] (version 1.94.0) on the processed and standardized protein intensities.

Protein groups were statistically analyzed using pairwise, two-sided Student's t-tests on the processed protein intensities between the replicates of respective control and test conditions. Only proteins with valid values in the test conditions were considered for analysis. Log2 fold changes were derived as the tested difference of means and the resulting p-values were adjusted for the false discovery rate (FDR) using the Benjamini-Hochberg procedure.

Using the results of our co-IP mass spectrometry data analysis we define robust interactors of p62 per treatment background as those proteins, whose intensities differ significantly (FDR < 0.05) between the p62 pulldown in a specific cell line and all other groups of IgG pulldown controls individually. Furthermore, to be able to determine the exclusivity of p62 interactors between each karyotype, we tested whether a protein could be either classified as an interactor using the conservative definition above or be excluded as an interactor by not meeting the more inclusive condition of a significant difference in intensities between the p62 pulldown in a cell line and only the matching IgG control with a p-value < 0.05. Subsequently, we only considered candidate interactors that fulfill this requirement for downstream analysis.

The results from the proximity biotinylation experiments were adjusted to account for alterations in overall protein levels across whole-cell extracts. We used the comprehensive proteomic dataset for HCT116 5/4 cells[8], while data obtained in this work were used for HCT116 13/3. Subsequently, the Student's t-tests were re-evaluated using the global log2 fold changes as the true difference in revised

means of the new null hypothesis. The proxitome fold changes were corrected by subtracting the global log2 fold changes. The analyses reflect whether the difference in protein intensities relative to the wild type is significantly higher within the p62 proximity than in the global abundance.

## Over-representation analysis

Hypergeometric tests were performed to assess the statistical significance of the overlap between the sets of differentially abundant proteins and the different molecular signature sets given all robustly identified proteins as a background. FDR-adjustment of $p$-values was done according to Benjamini-Hochberg within signature set categories. When indicated, the redundancy between related sets was reduced by applying the affinity propagation algorithm with negative log-transformed $p$-values for scoring to cluster similar sets and extract representatives as implemented in the WebGestalt R package[78] (version 0.4.6).

## Hydrophobicity scores

The function scaledHydropathyLocal implemented in R package idpr[79] (version 1.12.0) was applied to calculate the scaled hydropathy of all amino acid sequences in the UniProt database (UP000005640_9606) using a sliding window of 21 amino acids. Hydrophobicity of a protein group identified in the mass spectrometry proteomics experiments were then derived as the average maximum scaled hydropathy of all proteins in a group.

## Analysis of external cancer cell line multi-omics data

Cancer cell line data from the DepMap database was filtered to only include cancer cell lines that did not undergo whole genome doubling (WGD) as previously inferred[38]. This was done to exclude any confounding effects of WGD-positive cancer cell lines and to make the selection of cancer cell lines more comparable to WGD-negative HCT116. This left 127 cancer cell lines with both aneuploidy scores and transcriptome, proteome, and gene dependency data. Subsequently, cell line gene expression and dependency values were correlated with cell line aneuploidy scores using Spearman's rank correlation coefficient as implemented in the base R-package.

## Visualization of omics data

The results of the computational analysis of mass spectrometry data and the analysis of external cancer cell line multi-omics data were visualized using the ggplot2[80] (version 3.5.0), ggpubr (version 0.6.0), ggh4x (version 0.2.8) and cowplot (version 1.1.3) R packages.

## Cell lysis, SDS-PAGE and Western blot

Whole-cell lysates were prepared by solubilizing cell pellets in RIPA buffer supplemented with Protease and Phosphatase inhibitors (Merck) and vortexing every 5 min for 30 min at 4 °C. Samples were centrifuged at 2000 g for 10 mins to obtain the protein supernatant from which protein concentration was determined by the Bradford assay. To prepare samples for SDS-PAGE, lysates were diluted to 1 μg/μl with Laemmli buffer and boiled for 5 min at 95 °C. 10 μg of total protein was run on SDS-PAGE gels using either the Precision Plus Protein All Blue Standard (Bio-Rad) or Color Prestained Protein Standard (New England Biolabs) as marker. Proteins were transferred onto nitrocellulose membranes (AmershamProtran Premium0.45 NC, GE Healthcare Life Sciences, Sunnyvale, USA) by wet transfer at 100 V for 1 h and blocked in 3% BSA solution for 1 h at RT. Primary and secondary antibody incubations were carried out overnight at 4 °C and 1 h at RT, respectively. Protein signals were visualized by chemiluminescence and imaged on an Azure c500 system (Azure Biosystems, Dublin, USA). Quantification of protein bands was done in ImageJ. A list of antibodies used can be found in Supplementary Table 2.

## Immunostaining and fluorescence microscopy

Cells were grown on poly-L-lysine-coated coverslips prior to the indicated treatments. Thereafter, cells were washed three times using PBS and fixed using a 2% PFA/2% sucrose solution for 10 min at RT. Fixed cells were then washed three times using PBS, permeabilized using 0.1% TX-100 and blocked with 3% BSA for 1 h at RT. Primary and secondary antibodies (Supplementary Table 2) were incubated overnight at 4 °C, followed by 1 h at RT in the dark, respectively. In some instances, mitochondria were stained with MitoTracker Red CMXRos (Invitrogen, #M7512) according to the manufacturer's instructions. Coverslips were mounted on microscopy slides using VECTASHIELD® Hardset Antifade Mounting Medium containing DAPI (Vector Laboratories Wertheim, Germany). Slides were imaged using a semi-automated Zeiss AxioObserver Z1 (Oberkochen, Germany) equipped with an ASI MS-2000 stage (Applied Scientific Instrumentation, Eugene, USA), a CSU-X1 spinning disk confocal head (Yokogawa), a Laser Stack with selectable lasers (Intelligent Imaging Innovations, Denver, USA) and the Cool-Snap HQ camera (Roper Scientific). 40x air or 63x oil objectives were used under the control of the Slidebook6 software (Intelligent Imaging Innovations, Denver, CO).

## Analyses of protein aggregates

Protein aggregates were analyzed by staining of aggresomes using a commercially available aggresome detection kit consisting of a red fluorescent molecular rotor dye (Abcam, #ab139486). A positive control in which parental cells were treated with 10 μM MG132 for 6 h to induce aggresome formation was included according to the manufacturer's instructions. Briefly, a 10x assay buffer was diluted to 1x using deionized water. Fixed and permeabilized cells were incubated with 2 ml 1x assay buffer containing 1 μl of aggresome detection reagent and 2 μl of DAPI nuclear stain in a 12-well format, at RT for 30 min in the dark. Aggresome dye intensity was immediately thereafter analyzed by fluorescence microscopy as described above.

## Microscopy image analyses

p62-positive foci number and size were analyzed using Z-projections of p62-immunostained cells. Cell boundaries (i.e., regions of interest or ROI) were set manually by applying adjustments in brightness and contrast to the images in p62 channel. A threshold was set to reduce signal-to-noise ratio by using background signal from the WT cells as a reference. Additionally, a second threshold was set using the "Analyze particles" function in ImageJ to exclude structures less than 0.01 μm in size as p62-positive foci. Foci were then evaluated using an in-house automated pipeline developed based on ImageJ macros language. To account for differences in cell sizes, foci counts were normalized to the cells surface area estimated from the ROIs.

Colocalization of p62 and TOMM20 was estimated using the EzColocalization plugin in ImageJ[81]. A maximal intensity projection on the $Z$ axis was performed and this single in-focus field projection was used for the colocalization analysis. In the EzColocalization plugin, an automated threshold was set for the TOMM20 signal, whilst threshold of the p62 channel was set using background signal from the p62 channel in WT cells. Individual cells were outlined and saved to the ROI manager tool in ImageJ and analyzed with the "ROI" cell identification input. The output data was selected as Pearson correlation coefficient. This value ranges between -1 and 1. A perfect colocalization event is signified by a perfect linear correlation of fluorescence intensities of two images and assumes a Pearson correlation coefficient value of 1. A negative linear correlation results in a value of -1, and no correlation results in a value of 0[82].

To evaluate mitochondrial network, the Mitochondria Analyzer plugin[83] was used in ImageJ. The threshold algorithm of this plugin requires the empirical determination of block size and $C$-value parameters by a user, which prevents the merging of close but physically separate individual mitochondria as well as fake splitting

of continuous mitochondria. An optimal threshold for the TOMM20 channel was determined to be $C$ value of $1\,\mu m$ and block size of $1.45\,\mu m$, which gave the most accurate separation of the mitochondrial network in all cell lines used. Individual non-overlapping TOMM20 immunostained cells were then cropped out following the excision of mitochondria from adjacent cells when necessary. Mitochondria morphology analysis was performed on "Per-cell basis" and mean values of the following parameters were obtained: Branches, Total Branch Length, Branch Junctions, Branch Points, Mitochondrial Volume, and Mitochondrial Surface Area. Classification of network as extended, intermediate and perinuclear was determined using the number of branch junctions. Aggresome dye intensity was quantified in ImageJ using an in-house developed pipeline. The number of cells in each microscopy image was determined using maximal intensity projections along the $Z$ axis in the gray channel. Aggresome fluorescence intensity was quantified per image and normalized to the number of cells.

### Flow cytometry
Cultured cells were stained simultaneously with MitoTracker Green FM (Invitrogen, #M7514) and Deep Red FM (Invitrogen, #M22426) dyes according to the manufacturer's instructions to analyze mitochondria mass and membrane potential, respectively, and incubated for 20 min at $37\,°C$ and 5% $CO_2$. Unstained cells were included for background control. After incubation, cells were washed five times with PBS before harvesting by trypsinization. Cells were then resuspended in PBS and analyzed on an Attune NxT acoustic focusing cytometer (Thermo Fisher). The gating, data analysis, and visualization were performed using the FlowJo software (version 10).

### Florescence plate reader measurements
Reactive oxygen species levels were measured using CellROX Green reagent (Invitrogen, #C10444), according to the manufacturer's instructions. Briefly, cells were seeded in a 96-well format and incubated under cell culture conditions for 24 h. Subsequently, cells were washed thrice with 1x PBS and incubated with Fluorobrite DMEM (Gibco, #A1896701) containing 5 $\mu M$ CellROX Green and Hoechst 3342 dye for 30 min at $37\,°C$ and 5% $CO_2$. Fluorescence intensities were measured using GloMax Explorer Multimode Microplate Reader (Promega, Walldorf, Germany). CellROX Green intensity was normalized with Hoechst 3342 intensity.

### Mitochondria isolation and import assay
Isolation of mitochondria and protein import assay were performed as previously described[61]. Briefly, confluent cells were harvested and washed with ice-cold 1x PBS. Cells were then resuspended in homogenization buffer [220 mM mannitol, 70 mM sucrose, 5 mM HEPES-KOH pH 7.4, 1 mM EGTA-KOH pH 7.4] containing 1x Complete Protease Inhibitor Cocktail (Roche). and homogenized using a PTFE-based homogenizer (VWR). Homogenate was clarified by centrifugation at 600 g for 5 min at 4 °C to remove cell debris and nuclei. The supernatant was further centrifuged at 8000 g for 10 min at 4 °C to obtain the crude mitochondrial fraction, which was washed once in homogenization buffer before the import assay. Subsequently, import assay was performed with 20 $\mu g$ of mitochondria per reaction. Radiolabeled precursor proteins were synthesized using the SP6 promoter TNT Quick Coupled Transcription/Translation System (Promega) containing 20 $\mu Ci$ [$^{35}S$]-methionine at 30 °C for 1 h. The import reaction was started by incubating the crude mitochondria with cytosolic extracts of reticulocyte cells containing the radiolabeled precursor proteins at 30 °C, in the presence or absence of 1 mM CCCP. Precursor protein import was then stopped after 1, 2, 5, 10 or 15 min by placing mitochondria on ice. Proteinase K treatment was performed to degrade unimported substrates. The mitochondria were then washed in homogenization buffer containing 1 mM PMSF,

resuspended in Laemmli buffer and analyzed by SDS–PAGE followed by autoradiography.

### Mitochondrial oxygen consumption assay
100 $\mu g$ of isolated intact mitochondria in cytosolic extracts of reticulocyte cells containing 0.6 M sorbitol, 20 mM HEPES/KOH (pH 7.4) and 1 mM MgCl2 was used for oxygen consumption assay. The mitochondrial oxygen consumption rate was measured using a Clark-type oxygen electrode connected to a computer-operated Oxygraph control unit (Hansatech Instruments, Germany) at 25 °C with continuous stirring. The upper (100%) and lower (0%) limit of dissolved oxygen was calibrated using distilled water and sodium dithionite, respectively. Mitochondria were added to the chamber of the unit and respiration was initiated by the addition of 7 mM NADH. Data was recorded at intervals of 1 s (Oxygraph Plus software, Hansatech Instruments). In a control reaction, isolated mitochondria were treated with 10 mM CCCP for 15 mins to collapse the membrane potential before oxygen consumption measurements. Oxygen consumption (in $\mu mol/\mu l$) was normalized to the mitochondrial protein amount determined by Bradford assay.

### Quantification of mitochondrial DNA copy number by qPCR
Total DNA was extracted from cultured cells using the QIAamp DNA Mini Kit according to the manufacturer's protocol. Mitochondrial DNA was quantified from 10 ng of total DNA using SYBRGreen Mastermix and primer sets that amplify the non-coding D-Loop region and coding sequences of at least one respiratory chain complex subunit (Supplementary Table 3). The nuclear encoded $\beta_2$ microglobulin gene was used for normalization. Samples were assayed on the C1000 Touch Thermal Cycler (Bio-Rad) using the following run conditions: initial denaturation for 5 min at 95 °C followed by 45 cycles of 5 s at 95 °C, 15 s at 55 °C, 15 s at 72 °C and 1 s at 78 °C.

### Statistical analyses
Outside the analysis of omics data, GraphPad Prism 9 was used for statistical analysis. For each data set, the Kolmogorov-Smirnov-test was applied to determine whether variables follow a normal distribution. If yes, a two-tailed unpaired Student's $t$-test was used to compare the means of two groups. If no, a two-tailed $t$-test with Welch's correction was applied. For multiple comparisons, ordinary one-way ANOVA was used followed by Sidak's post hoc test. For non-normally distributed data, one-way ANOVA on ranks (Kruskal-Wallis H Test) was performed followed by Dunn's post hoc test. $P$-values in Supplementary Data 1-3 are derived from two-sided Student's $t$-test.

### Reporting summary
Further information on research design is available in the Nature Portfolio Reporting Summary linked to this article.

## Data availability
The mass spectrometry proteomics data (see also Supplementary Data 1–3) have been deposited to the ProteomeXchange Consortium via the PRIDE partner repository with the following identifiers: PXD052623 (APEX2-p62 global proxitome proteomics with WT, 13/3, 5/4), PXD061687 (APEX2-p62 global proxitome proteomics with WT and 3/3), PXD061712 (APEX2-p62 autophagosome lumen profiling proteomics), PXD052174 (p62 IP-MS), PXD061691 (p62 overexpression IP-MS), and PXD052637 (TMT-based global proteomics). Multi-omics data of CCLE cancer cell lines, including aneuploidy scores[36], RNAseq transcriptomics[38], mass spectrometry proteomics[39] and gene dependency[37] data were downloaded through the DepMap data portal[40] (version 22Q4). Gene Ontology Cellular Compartments (GOCC) and other gene sets were downloaded from the Molecular Signature database[84] (version 2022.1). Information on macromolecular complexes were downloaded from the CORUM database[85]. A list of 148

reported p62 autophagosomal lumen cargo proteins were taken from the supplementary data of Zellner et al.[42]. Source data are provided with this paper.

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

## Acknowledgements

We thank Ulrich Hartl, Len Neckers, Bianca Brundel and Anne Simonson for the plasmids. We thank Devi Bala Murugan, Caroline Erler, Zahra Pourmanouchehri, and Lena Johann for their help with the experiments. This project was supported by grants from the German Research Foundation (DFG) (GRK2737-STRESSistance to Z.S. and J.M.H., and Walter Benjamin Programme Award – Project number 510268075 – to P.S.A.), and the Landesforschungsinitiative Rheinland-Pfalz BioComp (to Z.S and J.M.H.). P.S.A. is additionally supported by the TU Nachwuchsring (RPTU Kaiserslautern) and an Add-on fellowship from the Joachim Herz Stiftung.

## Author contributions

P.S.A. performed the biochemical, proteomics and cell biology experiments. O.K., A.M, D.O.V.V. and C.H. contributed to the biochemical and cell biology experiments. P.S.A. and M.R. carried out the mass spectrometry analysis; J-E.B. and M.R. performed the analysis of the proteomics data, and J-E.B. performed the analysis of CCLE data. S.L. carried out the protein import experiments. Z.S., P.S.A., and J.M.H. supervised the study. P.S.A., J-E.B., and Z.S. wrote the manuscript; all authors critically reviewed and commented on the manuscript.

## Funding

## Competing interests

The authors declare no competing interests.
