## [Transparent Peer Review file · Nature Communications]

Aneuploidy-induced proteostasis disruption impairs mitochondrial functions and mediates aggregation of mitochondrial precursor proteins through SQSTM1/p62

Corresponding Author: Professor Zuzana Storchova

Version 0:

Reviewer comments:

Reviewer #1

(Remarks to the Author)

Amponsah et al. created a panel of polysomic cell lines and noticed that p62 level correlate with the degree of protein overexpression. They then perform p62 coIP MS and BioID and find that p62 interactions in polysomic cells are enriched for mitochondrial proteins. They go on to propose that protein overexpression causes mitochondrial defects which result in sequestration of mitochondria in p62 clusters destined for autophagy. The overall concept and its relation to extra-chromosomal defects and genomically unstable cancer cells is interesting, but several major weaknesses in the data undermine confidence in their model, as outlined below.

Major Critiques

- A major concern relates to the reliability of the data when the bulk of mechanistic analysis relied on two polysomic cell lines that in some cases produced different results. It's tough to make convincing conclusions based on a limited data set like this. For example, they detected different p62-associated mitochondrial proteins in the different polysomic cell lines. If it's a general defect in mitochondria, this doesn't make sense. How do they explain these differences? In another perhaps more important example, the fact that chaperones can rescue the defect in one polysomic cell line (3/3) but not another (5/4) raises doubt about the overall conclusion. To convincingly conclude that it's related to the degree of protein overexpression would need further evidence. These examples and others throughout the text raise concern that the observations could be related to clonal differences between lines or at least that the model is not accurate outside these particular lines.
- Another major concern relates to the overall conclusion that defects observed were due to general protein overexpression, implying that it's not due to expression of specific genes on the extra chromosomes. Again, relying mainly on the the 3/3 and 5/4 lines for many analyses makes this a tough sell. Could they use an orthologous approach (maybe a reductionist simple overexpression of protein X experiment) to show that they mito defects etc. are recapitulated by other overexpression systems?
- The enrichment of mitochondrial proteins in the polysomic cell lines is not convincing, except for 5/4 which is overexpressing p62 from the extra chromosomes. This raises suspicion that the overall model is not a widely applicable rule.
- The fact that they do not observe an enrichment in mito proteins in the p62 proxitome autophagosome experiment is very puzzling and no satisfying answer is given. Based on our understanding of p62, it's hard to imagine how clusters of p62 associated with mitochondrial proteins don't make it to the autophagosome. Like other aspects mentioned above, this point raises concern about the overall model and is mostly hand waived off in the text
- For the BioID data, it's critical that WT cells are used in the proteomics for comparison. In the results, it says that only 3/3 and 5/4 expressed APEX2, but then later it mentions that "The fold changes in p62-proximal proteins (hereafter called proxitome) in the polysomic cells relative to the parental diploid were then statistically tested against the corresponding fold changes in global protein abundance". This was confusing and it wasn't clear to me how WT cells were used.
- I don't see convincing differences in the blots in 7b. This is a key point in the paper and undermines the model.

Minor Critiques

- Figure legends need some additional details to explain the figures. For example, description of replicates and statistics
- No WT comparison in 1C so it makes comparisons difficult.
- What are the gray boxes in extended figure 1b?

- 2d and extended 1d have some karyotypes but different values. These panels weren't clear. What are they looking at?
- 4d not labeled as panel d (missing label?)
- They mention SOD1 import into mitochondria in the text but then show SOD2 in extended data. Was SOD1 a typo? The rationale for SOD1 there wasn't clear as it's mostly a cytosolic protein.
- Extended figure 2c was cited in the third paragraph of the results section, but it's not clear to this reviewer what this panel has to do with the text at that point. Did the authors mean to cite another panel?
- They didn't see extrachromosome-encoded proteins associating with p62. Can they at least confirm that these proteins are overexpressed? It's also possible that MS missed them or that they're in the p62 condensates but not abundant enough or interacting strong enough to be pulled out. Their premise is that the extra protein load causes p62 condensate build-up so this would be important.

Reviewer #2

(Remarks to the Author)

In this manuscript, Amponsah et al demonstrated that proteotoxic stress in polysomy cells could compromise mitochondrial functions by sequestration of mitochondrial precursor proteins in cytosolic p62-bodies. More specifically, the authors developed unique polysomic models by chromosome transfer, which showed cytosolic p62-bodies abundantly. In addition, mitochondrial proteins were enriched within the cytosolic p62 interactome and proxitome in the polysomic cells, suggesting mitochondrial proteins would be trapped in p62-bodies to inhibit trafficking of these proteins into mitochondria. Finally, the authors demonstrated that mitochondrial functions were impaired in the polysomic, showing reduced oxygen consumption or mitochondria DNA abundance. Protein chemistries to identify mitochondrial proteins in p62-bodies were great. These findings should provide new insights on aneuploid biology, filling a functional gap between proteotoxic stress and metabolic stress. However, there are couples of major issues to be cleared: 1) how and why p62 bodies were profoundly formed in the polysomic cells? 2) Were mitochondria in polysomic cells physiologically damaged? 3) How does gain of chromosome(s) affect metabolic networks in the polysomic models. Major concerns are concretely described below.

Major concerns

(1) Effect of aneuploid-mediated proteotoxic stress on p62-body in polysomic cells

While the authors showed increased p62-bodies in the polysomic cells, it is unclear whether aggregated proteins were accumulated in the polysomic cells to trigger p62-body formation. Proteotoxic stresses, such as activation of UPR signaling for ER stress, need to be determined in the polysomic cell lines compared to their parental cells.

(2) Effects of overexpression of chaperon protein on p62-body

The authors showed that overexpression of chaperon proteins reduced numbers of p62-bodies in the polysomic cells, while the size of p62-bodies turned to be larger chunks (Fig. 6i), suggesting that aggregated proteins may be more accumulated in these cells. Reversely, if overexpression of chaperon proteins cleared the protein aggregation, aggregated proteins should not promote p62-body formation in the polysomic cells. The authors need to explain how overexpression of chaperon proteins affected aggregated proteins and larger chunks of p62 bodies. Rescue effects of overexpression of chaperon proteins on mitochondria activities may be determined as well.

(3) Mitochondria morphology in the polysomic cells

The authors showed that polysomic cells reduced import of mitochondria proteins and accumulated mitochondria proteins in the cytosol, suggesting that mitochondria in these cells may be aberrant due to lack of their components. However, it is unclear whether mitochondria in the polysomic cells are physiologically abnormal. The authors should show the morphological changes of mitochondria, such as damaged mitochondria, in the polysomic cells. Electric microscopy may be helpful. Also is the expression of mitochondria-associated proteins comparable in parental and polysomic cells?

(4) Effects of chromosome gain on metabolic networks

The authors showed polysomic cells compromised mitochondria function by sequestration of mitochondria proteins in p62-bodies. Mitochondria disfunction may change metabolic networks in the polysomic cells presumably to shift TCA cycle to glycolysis. The authors should show the effects of mitochondria disfunction on the metabolic networks in the polysomic cells.

(5) Aneuploidy-related p62 protein expression and autophagy flux

Fig1b: p62 protein expression changes in polysomic cells are shown.

However, no data show the autophagy flux state; please show the flux for LC3 (I and II) (on the same membrane, if possible).

(6) Nuclear localization of p62 in the polysomic cells

HADHB (mitochondrial hydroxy acyl-CoA dehydrogenase trifunctional multienzyme complex subunits) and p62 showed nuclear localization in polysomic cells. Do proteins co-localizing with nuclear p62 exhibit normal autophagic flux? The authors should confirm normal autophagy flux by an autophagy flux assay with a fluorescent probe such as "tandem fluorescent protein-tagged LC3" or "GFP-LC3-RFP-LC3ΔG". The authors also need to compare and quantify the co-localization rate of p62 on autophagosomes, which may be functionally intact with the cytoplasmic localization.

Other Concerns

Fig2C, Extended Data Fig1: The expression level of mitochondria-associated proteins appears to decrease. On the other hand, in Fig. 7A, the expression levels of HADHA and HADHB are unchanged (precursor is increased).

Fig2C: Present a Fig for each fluorescence channel, showing the change in fluorescence intensity of HADHB.

Fig2C, Extended Data Fig1: Do HADHB and TOMM20 expression co-localize?

Fig4:

> Surprisingly, we found no over-representation of mitochondrial proteins in the polysomy-specific p62 cargo in autophagosomal lumen (Fig. 4c, d, Extended Data Fig. 3c).

There is no "d" in Fig. 4.

Reviewer #3

(Remarks to the Author)

Aneuploidy affects cellular physiology and development. Previous studies suggested that increased expression of surplus proteins encoded on the extra chromosomes may contribute to the abnormalities. This results in proteostatic crisis as manifested by increased recruitment of aggregated proteins to p62, a key player in aggrephagy and other types of autophagic activities. However, it remains unclear whether this affects the homeostasis of specific groups of proteins, followed by specific cellular defects in aneuploidy cells. In the current manuscript, the authors addressed this gap of knowledge.

The authors elegantly generated cell lines that contain extra copy (or copies) of chromosomes 3, 5, 13 and 21. They found that the formation of p62 foci correlates with the loading of surplus protein coding genes in these polysomic cells. Proteomic analysis of p62 interactors using immunoprecipitation and proximity labeling assay showed enrichment of many mitochondrial proteins, but not those encoded on the extra chromosomes in p62 bodies in the cytosol. The authors also showed that cells with extra chromosomes contain stressed and seemingly less functional mitochondria, with reduced levels of mitochondrial proteins and mtDNA. Interestingly, the polysomic cells appear to accumulate unimported mitochondrial precursor proteins. In organello protein import assay also showed that The overall mitochondrial protein import capacity is also reduced in polysomic cells. Based on these observations, it was concluded that sequestration of mitochondrial proteins by cytosolic p62 bodies may impairs mitochondrial health. This may contribute to cellular stress in polysomic cells.

This is an interesting study as it uncovered the mitochondrial protein import process as a vulnerable process affected by polysomy. The authors provided compelling evidence that mitochondrial precursors are sequestered by p62 in aneuploidy cells, supported by solid data from two independent proteomic approaches. This is an important step towards the better understanding of how aneuploidy affects cellular proteostasis. Strong data also comes from the experiments showing that mitochondrial protein import is affected and that unimported mitochondrial precursor proteins accumulate in polysomic cells. There are a few weaknesses that damper the overall enthusiasm which need to be addressed.

Major weaknesses:

1. One of the key observations is that the formation of p62 foci correlates with the loading of surplus protein coding genes in these polysomic cells. As noted by the authors, the p62 gene is present on chromosome 5. This unfortunately confounds the quantification of p62 bodies in the 5/3 and 5/4 cell lines, as the increase of p62 bodies may also result directly from elevated p62 copy number. The data in the 13/3 cell line is convincing, showing the sequestration of mitochondrial proteins. Perhaps, a negative control cell line with increased p62 expression may be used to validate the data in 5/4 cells.
2. The authors showed changes to mitochondrial localization, reduced mtDNA copy number and OXPHOS components in polysomic cells (Fig. 5). This could just reflect a general reduction of mitochondrial biogenesis in response to global proteostatic stress, instead of a specific functional impairment and perturbed mitochondrial genome maintenance as claimed in the manuscript. On the other hand, the data with mitochondrial functional assay are problematic. First, no details were provided regarding how the experiments were executed, except that a reference is provided in which isolated mitochondria from yeast cells were analyzed. How much isolated mitochondria are used in the assays? I am afraid that the scale on the Y axis may be wrong (Extended Data Fig. 6b). It is unclear why NADH is used as a substrate, instead of using the standard substrates glutamate/malate. Mammalian mitochondria do not seem to directly oxidize NADH from the cytosolic side. Controls for standard assays are missing, including oligomycin-inhibited and FCCP-stimulated respiration, making it difficult to evaluate whether the isolated mitochondria are intact. The data on Opa1 processing and Drp1 levels provide little information regarding mitochondrial functionality (Extended Figure 6d-e). It seems to me that the authors try to push the idea that sequestration of mitochondrial precursor proteins by p62 causes mitochondrial dysfunction. This would be fine given the drastic reduction of OXPHOS components in the polysomic cells, as long as the experiments should be done correctly. But on the other hand, mitochondrial bioenergetics is very flexible in cell lines. Mitochondrial protein import stress can affect cell physiology in many other ways, without necessarily affecting respiration in a major way.
3. Figure 6 – The authors measured p62 body density and size in polysomic cells overexpressing several chaperones and heat shock transcriptional factor. Based on the reduction of p62 foci number, the authors concluded that "...the proteotoxic stress and mitochondrial defect can be rescued by overexpression of cytosolic chaperones...". And that "Our data suggests that impaired protein folding and chronic overexpression of extra proteins in polysomic cells facilitate formation of cytosolic p62 bodies that sequester mitochondrial proteins, thus impairing mitochondrial functions". These conclusions are poorly supported by the data. The p62 foci are clearly larger in cells overexpressing the chaperones. This may also be interpreted as increased aggregation propensity, instead of alleviating proteostatic stress. It may be an oversimplification in that expressing one single chaperone can fix the problem, especially no supporting data are available showing that mitochondrial dysfunction is rescued by the chaperones. Given that mitochondrial protein overexpression and unbalanced protein loading/import are known to directly cause proteostatic stress in the cytosol, it is possible that disturbed cytosolic proteostasis by polysomy may directly disrupt mitochondrial precursor delivery to mitochondria and the formation of p62 foci

may just be a stress response in the cytosol. The authors need to discuss this in the manuscript.

4. It is also misleading when stating that “.....the impaired protein folding and imbalanced protein homeostasis due to aneuploidy initiates protein aggregation in cytosolic p62 bodies, which then sequester mitochondrial proteins before they can be imported into mitochondria”. p62 normally sequesters ubiquitinated proteins in aggregates. It is unlikely that these proteins readily regain the unfolded state for further import.

Minor comments:

1. Figure 1 - Figure 1c, which chromosome encodes beta-actin should be indicated in the legend. In the text - “..... demonstrating normal autophagic flux...”. As no rate of flux is measured, “normal” is unjustified. Figure 1D – The images lack the delineation of cell boundary, making it difficult to ascertain whether the green puncta come from a single cell. Figure 1h – 1i, add p62 to the Y axis.
2. Figure 3d – please check whether the p values are correct.
3. Extended Data Fig. 6b – “5/4” instead of “54”
4. The authors may consider moving Extended data Fig. 1b to the main figure, given the importance of the data for supporting the overall conclusion.
5. Figure 7b-d – what does “20%” denote?
6. Ref. 37 is incomplete.
7. Page 11, line 15: broken sentence.
8. Page 11, “mitochondrial internal membrane-localized”. I guess it meant “...inner membrane...”.
9. Page 11, line 15, broken sentence.

Reviewer #4

(Remarks to the Author)

"I co-reviewed this manuscript with one of the reviewers who provided the listed reports. This is part of the Nature Communications initiative to facilitate training in peer review and to provide appropriate recognition for Early Career Researchers who co-review manuscripts."

Version 1:

Reviewer comments:

Reviewer #1

(Remarks to the Author)

This is an interesting study and I think the overall manuscript and presentation of data have improved significantly from the first version. The authors addressed my critiques. I have no further concerns.

Reviewer #2

(Remarks to the Author)

Overall Evaluation

The authors have made substantial improvements to the manuscript by incorporating additional experimental data. However, several critical concerns previously raised remain insufficiently addressed. The major unresolved issues include: (1) the correlation between p62 accumulation/foci formation and activation of the unfolded protein response (UPR), (2) the interpretation that increased p62-positive body size contributes to reduced cellular toxicity, (3) the physiological impact on mitochondrial function resulting from reduced mitochondrial components, and (4) overall experimental quality control. These issues must be rigorously and comprehensively addressed prior to publication.

1) Correlation between p62 accumulation/foci formation and stress response intensity in polysomic cell models

While numerous studies have reported that aneuploidy induces proteotoxic stress, I strongly encourage the authors to directly assess the activation of proteotoxic stress-related signaling pathways in the polysomic cell models utilized in this study, for the following reasons.

First, the current analysis relies solely on p62 as a marker of proteotoxic stress (Fig. 1b–c). If impaired protein homeostasis indeed leads to p62 accumulation and foci formation, it is expected that additional components of proteotoxic stress pathways—particularly those involved in the unfolded protein response (UPR)—would be similarly modulated. The inclusion of multiple, well-established markers would provide a more comprehensive and quantitative assessment of stress response dynamics.

Second, the relationship between the burden of surplus proteins—presumably correlated with the size of the additional chromosome—and the magnitude of stress responses remains unclear. The authors report that p62 expression and foci formation increase in proportion to the size of the supernumerary chromosome. This observation suggests that other proteotoxic stress markers may also be quantitatively modulated in a similar manner. To substantiate the interpretation that p62 dynamics reflect a broader stress response, the authors should evaluate and present correlations between chromosomal burden and changes in multiple proteotoxic stress markers.

These additional analyses would significantly strengthen the manuscript by providing a more rigorous and holistic understanding of stress response mechanisms in aneuploid cells.

2) Interpretation of increased p62-positive body size as a mechanism of reduced cellular toxicity

The authors propose that the formation of fewer, larger p62-positive bodies contributes to reduced cellular toxicity, citing previous literature suggesting that smaller, dispersed aggregates may be more toxic than larger ones. While this hypothesis is plausible, it remains speculative within the context of the current study due to the lack of direct supporting evidence. To validate this interpretation, the authors should provide experimental data demonstrating that cells harboring larger p62-positive bodies exhibit reduced markers of cellular stress and/or improved viability compared to those with numerous smaller aggregates. For instance, alterations in UPR markers or other stress indicators, as well as functional recovery—such as restored mitochondrial function—would offer more convincing support for the proposed protective role of large aggregates. In the absence of such data, the conclusion that aggregate size correlates with toxicity reduction remains insufficiently supported.

3) Physiological consequences of reduced mitochondrial components on mitochondrial function

Despite previous comments, the authors have not adequately addressed the potential consequences of reduced mitochondrial components on mitochondrial function. The current evidence, particularly the low-resolution TOMM20 immunofluorescence images (Fig. 5a, 5d), is insufficient to evaluate morphological or functional mitochondrial changes. To robustly assess mitochondrial dysfunction, the authors should employ additional, direct methodologies, such as: Transmission electron microscopy to evaluate detailed mitochondrial morphology, including fission/fusion states and cristae structure; Seahorse extracellular flux analysis to quantify oxygen consumption rates and mitochondrial respiratory capacity; TMRM staining to assess mitochondrial membrane potential and functional integrity. Given the claim of “compromised mitochondrial function” (line 37), the inclusion of such assays is essential for substantiating this interpretation. These additions would significantly enhance the credibility and impact of the mitochondrial-related findings in the manuscript.

4) Quality control and reproducibility of foundational data

While the authors repeatedly state in their rebuttal that “these results are already available” or that “these data have been published previously,” I must respectfully emphasize that inclusion of fundamental data supporting the core hypotheses is essential within the current manuscript, regardless of prior publications. The existence of similar data in earlier studies does not ensure reproducibility under the current experimental conditions, which may differ in model, methodology, or context. Although a full replication of previously reported experiments is not required, I strongly recommend that the authors include a minimal but critical subset of key data, reproduced within the present experimental framework, to confirm the validity of essential findings. Such confirmatory evidence is not only scientifically appropriate but also necessary to ensure adequate quality control and methodological transparency.

In summary, while the authors have made commendable efforts to improve the manuscript, the aforementioned concerns remain unresolved. Addressing these points through the incorporation of additional, rigorously conducted experiments will be essential for ensuring the scientific validity and integrity of the study prior to publication.

Reviewer #3

(Remarks to the Author)

The authors should double check the accuracy of the bioenergetic data as I pointed out in the review. It sounds like the isolated mitochondria are super powerful that can burn so much oxygen in the mM range per micro gram of isolated mitochondria. Also, please make sure that CCCP is used correctly that is 10 mM instead of 10 microM. Otherwise I do not have any other comments to add.

Reviewer #4

(Remarks to the Author)

Point-by-point response to reviewers' comments (responses are in blue):

Reviewer #1 (Remarks to the Author):

Amponsah et al. created a panel of polysomic cell lines and noticed that p62 level correlate with the degree of protein overexpression. They then perform p62 coIP MS and BioID and find that p62 interactions in polysomic cells are enriched for mitochondrial proteins. They go on to propose that protein overexpression causes mitochondrial defects which result in sequestration of mitochondria in p62 clusters destined for autophagy. The overall concept and its relation to extra-chromosomal defects and genomically unstable cancer cells is interesting, but several major weaknesses in the data undermine confidence in their model, as outlined below.

We are grateful to the reviewer for the interest in our work and for taking a critical look at our manuscript. Importantly, we did not conclude from our data that mitochondria are sequestered by p62, but mitochondrial proteins which were not yet imported into the organelle. This is evident from the proteomics data, which only shows a subset of the mitochondrial proteome bound to p62. We believe that better explanation and additional data will reliably address the remarks and criticisms.

Major Critiques

- A major concern relates to the reliability of the data when the bulk of mechanistic analysis relied on two polysomic cell lines that in some cases produced different results. It's tough to make convincing conclusions based on a limited data set like this.

We respectfully disagree with the reviewer on this point. The phenotypes describing accumulation of cytosolic p62-positive protein deposits, along with the related mitochondrial phenotypes, were assessed in a panel of five polysomic cell lines in addition to the parental diploid. The proteome analysis was performed in the parental cell line and two aneuploid cell lines with medium (trisomy of chromosome 13, 13/3) and highest (5/4) p62 accumulation for in-depth proteomic analysis. We now **added proxitome analysis of another cell line** (3/3). The analyses of mitochondrial phenotypes, as well as the interacting protein colocalization for validation, were performed in all six cell lines (see prev. Figs. 5, 7, Extended Data Figs. 5, 8, now Figs. 5, 7, Extended Data Fig. 7, 9). In only a few instances we chose to further analyze the mitochondrial phenotype in the cell line that showed the most severe defect (i.e., 5/4, prev. Extended Data Fig. 6, now Extended Data Fig. 7h-l). To better examine the difference between "severe" (5/4) and "mild" (3/3) proteotoxic stress and mitochondrial phenotypes, we now also added another cell line to chaperone overexpression experiments. The import assay into mitochondria was performed in the parental cell line and three aneuploids. It is therefore not correct to state that our data is not reliable because "the bulk of mechanistic analysis relied on two polysomic cell lines."

For example, they detected different p62- associated mitochondrial proteins in the different polysomic cell lines. If it's a general defect in mitochondria, this doesn't make sense. How do they explain these differences?

We consistently found that mitochondrial proteins were associated with p62 in polysomic cells. Moreover, many of the p62-bound proteins were abundant components of the mitochondrial proteome, in particular constituents of OXPHOS complexes or mitochondrial ribosomes. Thus, most of these proteins were constituents of heterooligomeric complexes, indicating a shared pattern across polysomic cells. However,

the individual proteins associated with p62 differed to some extent, potentially reflecting different degree of proteome imbalance. In cells with milder proteotoxic stress (13/3, 3/3), p62 proxitome is enriched for inner mitochondrial membrane and matrix proteins (previous Extended Data Fig. 2c, now Extended Data Fig. 2d). Supportive of this, we also observed increased enrichment of proteins that are predicted to possess matrix targeting sequences (MTS) specifically in 13/3 (previous Extended Data Fig. 2d, now Extended Data Fig. 2c). Proteins destined for the mitochondrial matrix and inner membrane are imported in unfolded form through the translocases of the outer and inner membranes of mitochondria. Under proteotoxic stress, when the import process is disrupted, precursor proteins accumulate in the cytosol and, as demonstrated by our study, become potent substrates for p62.

In contrast, 5/4 cells, which experience- greater proteotoxic stress, additionally show a significant accumulation of proteins of the outer mitochondrial membrane (OMM) in the p62 proxitome (previous Extended Data Fig. 2c, now Extended Data Fig. 2d). OMM proteins, which are generally less dependent on the complex translocation machinery, are inserted into the outer membrane via the TOM complex. In agreement, and in contrast to 13/3, we observe the equal representation of MTS and non-MTS proteins in the p62 proxitome of 5/4 cells (previous Extended Data Fig. 2d, now Extended Data Fig. 2c). The accumulation of these proteins in 5/4 cell line thus suggests defects in their proper insertion or stabilization at the outer membrane. This could lead to their sequestration by p62, which indicates that in cells with more severe proteotoxic stress, mitochondrial proteins that do not require matrix targeting and are localized at OMM might be primarily affected. The observed accumulation of hydrophobic proteins in 5/4 cells (Extended Data Fig. 2e) further supports the idea that mitochondrial dysfunction in 5/4 might stem stronger from misfolding or aggregation of membrane proteins, which tend to be hydrophobic. Altogether, our data suggests that different types of mitochondrial proteins are vulnerable at different thresholds of stress. Indeed, a comprehensive analyses of mitochondria protein uptake dynamics in HeLa cells revealed varying degree of import of different classes of mitochondrial proteins during stress, with the more hydrophobic mitochondrial inner membrane (such as OXPHOS) and mitoribosomal proteins being particularly affected (Schäfer et al. 2022, *Mol. Cell.*). Thus, the complex and multifaceted nature of mitochondrial protein import, which operates differently for distinct classes of proteins, should be considered when interpreting the findings. Future in-depth investigation is needed to precisely map the routes through which different classes of unfolded mitochondrial proteins of different mitochondrial compartments are affected in response to varying degree of proteotoxic stress. We added a remark to address this point in the Discussion.

In another perhaps more important example, the fact that chaperones can rescue the defect in one polysomic cell line (3/3) but not another (5/4) raises doubt about the overall conclusion. To convincingly conclude that it's related to the degree of protein overexpression would need further evidence.

We believe this is related to the degree of proteotoxic stress, as stated earlier. It is also important to point out that these cells have mild but chronic proteotoxic stress due to the overexpression of thousands of unneeded proteins. A transient overexpression of chaperones may not be enough to achieve an immediate rescue (within 48 h) of the mitochondrial phenotype in a cell line with “severe” stress (i.e., 5/4). Any result suggesting to the contrary would be surprising. We now analyzed an additional cell line, trisomy of chromosome 13, with an intermediate level of proteotoxic stress. This analysis revealed reduced p62 numbers and a rescue of the TOMM20 – p62 colocalization phenotype to a similar degree as observed in the cell line with trisomy of chromosome 3 (now Fig. 6f-h, Extended Data Fig. 8). We also show in the revised version that the overexpression of chaperones reduces formation of aggregates, as

evidenced by reduced staining with a fluorescent rotor dye in all polysomic cell lines (now Fig. 6i-k). Furthermore, our new results demonstrate that overexpression of p62 *per se* in the parental cells (even at the levels higher than in aneuploid cells, now Fig. 4c-d) is not sufficient to trigger the mitochondrial phenotypes that we identified in aneuploid cells (now Fig. 6a-e). In agreement, p62-IP MS in wild type cells overexpressing p62 did not show similar enrichment of mitochondrial proteins in p62-positive aggregates (now Fig. 4f). Thus, the strong enrichment of mitochondrial proteins in the p62-associated aneuploid subproteomes is not in response to the accumulation of p62 *per se* but requires the additional proteotoxic stress caused by aneuploidy. These new results strongly suggest that inability to rescue mitochondrial phenotypes in the 5/4 cell line, which carries the largest number of extra protein-encoding genes, is due to the highest level of proteotoxic stress in these cells.

These examples and others throughout the text raise concern that the observations could be related to clonal differences between lines or at least that the model is not accurate outside these particular lines.

We do not discount that karyotypic differences in the various cell lines could also influence our result. At the same time, we consider the evidence gained from three to five different polysomic cell lines, including the validations and analyses of mitochondrial phenotypes (e.g., mitochondrial network, mtDNA copy number, levels of mitochondrial proteins (Fig. 5) is sufficiently strong to support our claim that proteotoxic stress caused by extra chromosome leads to mitochondrial dysfunction.

- Another major concern relates to the overall conclusion that defects observed were due to general protein overexpression, implying that it's not due to expression of specific genes on the extra chromosomes.

Indeed, a plethora of this and other previously published results from several laboratories including ours indicates that the phenotypes in aneuploid cells arise not due to expression of specific genes, but due to a synergic effect of low overexpression of several hundreds of proteins. In cells with a single extra chromosome, the overexpression of individual proteins is at the level of 50 % at most (1.5x), and most of them also show so called "dosage compensation", where their abundance is lower than expected from chromosome copy number change (e.g., 1.3x) (e.g., Stingle et al, MolSysBiol, 2012, Schukken&Sheltzer, Genom Res 2022); such abundance usually does not lead to a strong effect. There is also an evidence convincingly concluding that the major phenotypes of cells with a single extra chromosome are not due to the 1.5x overexpression of one single protein, but due to a low overexpression of many different proteins (e.g., Bonney et al 2015, *Genes Dev*). Typically, a single protein has to be overexpressed 10 to 100 times to result in a phenotype (Sopko et al 2006, *Cell*, Geiler-Samerotte et al 2010, *PNAS*). It has been previously shown that even an overexpression of a single mitochondrial carrier protein induces formation of large cytosolic protein aggregates that sequester unimported mitochondrial proteins, but even here a threshold level of overexpression is required to achieve a phenotype (Liu et al. 2019, *MBoc*). Additionally, while individual proteins encoded on the aneuploid chromosomes may have specific effect on the phenotypes, the fact that various polysomic cell lines share a similar phenotype suggests that this is rather a general consequence of a chronic, albeit mild protein overexpression.

However, we agree that this is an important point. An effect of an individual protein overexpression should not be dismissed, and we will elaborate this point in the Discussion.

Again, relying mainly on the the 3/3 and 5/4 lines for many analyses makes this a tough sell.

We respectfully disagree with this assertion is inaccurate. Most analyses were performed in all six cell lines, only the proteomic analysis and some of the detailed phenotypic characterizations were done in a limited number of polysomic cell lines. We now added another cell line for the proteomic analysis as well as for the heat shock protein overexpression experiments; these new cell lines show comparable phenotypes in all analyzed aspects.

Could they use an orthologous approach (maybe a reductionist simple overexpression of protein X experiment) to show that they mito defects etc. are recapitulated by other overexpression systems?

We are grateful to the reviewer for this suggestion. We showed that overexpression of p62 alone in the parental diploid cell line does not induce mitochondrial defects (i.e., p62-TOMM20 colocalization; now Fig. 6a-e). In the revised version, we determined the p62 interactome of the parental diploid cells overexpressing p62 (7x more, in comparison, we observe approximately 5x higher levels of p62 in the cell line tetrasomic for chromosome 5, 5/4; now Fig. 4c-d). These new data show that p62 overexpression in the parental diploid cells is not sufficient to cause mitochondrial defects and that the interactome is not similarly enriched for mitochondrial proteins as observed in the polysomic cell lines. These new results are now part of the Figure 4c-f.

To independently mimic proteotoxic stress by another approach, we transiently overexpressed the aggregation-prone Huntingtin Q97-EGFP, and as a control its non-aggregating variant Q25-EGFP in the parental cell line. We observed a strong accumulation of cytosolic p62 bodies that also colocalized with the EGFP signal in Q97-EGFP transfected cells (Fig. R1a). Analysis of p62-TOMM20 colocalization also showed a marginal but statistically significant increase in Pearson correlation coefficient in Q97-EGFP-overexpressing cells (Fig.R1b-e). However, it is important to note that this is a transient overexpression system. In contrast, the established aneuploid cell lines used in our study, experience chronic proteotoxic stress, which likely accounts for the more pronounced mitochondrial defects observed in polysomic cells compared to those seen in the transient overexpression models. We also treated parental cells with tunicamycin, which triggers the unfolded protein response (UPR), and observed no effect on p62 deposits formation (data not shown). These findings suggest that protein aggregation stress only partially underlies the mitochondrial defects in polysomic cells, which likely requires a chronic proteotoxic stress environment to manifest a severe phenotype, and that UPR is not strongly involved. We prefer not to include this data in the manuscript, but can add it if requested.

- The enrichment of mitochondrial proteins in the polysomic cell lines is not convincing, except for 5/4 which is overexpressing p62 from the extra chromosomes. This raises suspicion that the overall model is not a widely applicable rule.

We respectfully disagree with this inaccurate interpretation of our data. Mitochondrial proteins are enriched in the proxitome in both the 13/3 and 5/4, as well as in the newly added 3/3 cell line, as shown in our manuscript (previous Figs. 2, 3, Extended Data Fig. 2, now Figs. 2, 4, Extended Data Figure 2). The interaction of p62 with mitochondrial proteins was confirmed by fluorescence microscopy in all other polysomic cell lines (previous Fig. 2c, now Extended Data Fig. 5b). The effect is stronger in the 5/4 cell line, which is the cell line with the most prominent proteotoxic stress and mitochondrial phenotype. As one can see in Fig. 1b-c, the abundance of p62 in cell line with trisomy of chromosome 3 is approximately 4x higher compared to wild type (note that this cell line has the standard number of chromosome 5), while in cell lines with four copies of chromosome 5 – and thus four copies of SQSTM1 gene – the increase is approximately 5-fold. Thus, the increase of p62 is not limited to the cells with extra copies of chromosome 5 or with the overexpression of p62 *per se* (now Fig. 6a-d). Additionally, our new experiment revealed that overexpression of p62 in the parental diploid cell line does not lead to an enrichment of mitochondrial proteins as observed in the polysomic cells (now Fig. 4c-f), supporting the fact that the observed phenotypes in the polysomic cells are not solely due to the p62 overexpression.

- The fact that they do not observe an enrichment in mito proteins in the p62 proxitome autophagosome experiment is very puzzling and no satisfying answer is given. Based on our understanding of p62, it's hard to imagine how clusters of p62 associated with mitochondrial proteins don't make it to the autophagosome. Like other aspects mentioned above, this point raises concern about the overall model and is mostly hand waived off in the text.

The function of p62 is not limited to autophagy. It bridges several pathways that ensure protein homeostasis in cells, including cytosolic aggregation, ubiquitin proteasome system (UPS), and autophagy, among others. p62 has been found in protein condensates/aggregates extruded to the cellular exterior in a mechanism termed as “secretory autophagy” (Leidal et al. 2020, *Nat. Cell. Biol.*; Solvik et al. 2022, *JCB*). In cells with saturated autophagy, mitochondrial homeostasis is maintained through exocytosis of whole mitochondria or mitochondrial material (Phinney et al. 2015, *Nat. Commun*; Choong et al. 2021, *Autophagy*; D’acunzo et al. 2021, *Sci. Adv.*). This can be the case in aneuploid cells, as acutely aneuploid cells exhibit saturated autophagy (Ohashi et al. 2015, *Nat. Commun*; Santaguida et al. 2015, *Genes Dev*), and constitutively aneuploid cell lines increase autophagy (Stingele et al. 2012, *Mol Syst Biol*; Krivega et al. 2021, *Commun Biol*). Our proteomic data also reveals enrichment of external encapsulating structure and extracellular matrix components with p62 in polysomic cells (previous Fig. 4c, now Fig. 3c), supporting the possibility of the extrusion of the material. Clearly, there are several routes involved in the channeling of mitochondrial p62 interactors. As we mentioned in our discussion, this is a subject of intense ongoing investigations and beyond the scope of the current manuscript.

It should also be noted that the involvement of p62 in mitophagy is complex. On the one hand, p62 has been suggested to function as an adaptor protein in both PINK/Parkin-dependent (Geisler et al. 2010, *Nat. Cell. Biol.*) and -independent (Yamada et al. 2018, *Cell Metab*; 2019, *Autophagy*) mitophagy. On the other hand, it also clusters damaged mitochondria without directly facilitating mitophagy, a function reserved only for OPTN, NDP52 (or CALCOCO2), and, to a lesser extent, TAX1BP1 (Narendra et al. 2010, *Autophagy*; Lazarou et al 2015, *Nature*; Nguyen et al. 2023, *Mol Cell*). These mitophagy adaptors were also identified in the autophagosomal lumen proximate of p62 (previous Fig. 4b, now Fig. 3b), which may well mediate mitophagy in the polysomic cells. Finally, p62 is proposed to regulate mitochondrial morphology through its colocalization with TOMM20 (Seibenhener et al. 2013, *Biochim Biophys Acta*), and we observed compromised mitochondrial morphology in the polysomic cell lines (Fig. 5d-e). These data and other previously published observations suggest several possible fates of p62-associated mitochondrial proteins. In the revised manuscript, we describe the multitude of the p62 functions.

- For the BioID data, it’s critical that WT cells are used in the proteomics for comparison. In the results, it says that only 3/3 and 5/4 expressed APEX2, but then later it mentions that “The fold changes in p62-proximal proteins (hereafter called proximate) in the polysomic cells relative to the parental diploid were then statistically tested against the corresponding fold changes in global protein abundance”. This was confusing and it wasn’t clear to me how WT cells were used.

We are sorry for this apparent confusion, however we had pointed out in the text and in the figures that wild type cells (WT) were used throughout our study for the comparison. The proteomic data shown in the manuscript are always normalized to the parental diploid WT and we made this clear in the text. Both the global protein abundance and the p62 proximity proteomics data were normalized to WT and then compared with each other. This way, we determined which proteins are enriched in the p62 proximate of the polysomic cell lines relative to the WT, while accounting for respective changes in the global protein abundance. This information was provided in the text, in the figures, and in the figure legends, and the tables also show the complete data sets, including the data set for the wild type.

- I don’t see convincing differences in the blots in 7b. This is a key point in the paper and undermines the model.

Previous Fig. 7b (now a part of the Fig. 7c) is a comparison of *in-organello* import between the WT and the 13/3 cell line that has milder mitochondrial defects. On this representative image, the difference is perhaps less visible, but quantification of all the bands reveals the difference-. The more clearly visible differences are in 3/3 and 5/4 cell lines (previous Figs. 7c-d, now a part of the Fig. 7c), which have stronger proteotoxic stress. The observation in the Fig.7c (previous Fig. 7b) does not undermine our model, it rather supports it. The cell lines represent a physiological model of proteotoxic stress with different levels of stress (evident by p62-positive body formation, Fig. 1) scaling with the number of surplus protein-coding genes on the polysomic chromosome.

Minor Critiques

- Figure legends need some additional details to explain the figures. For example, description of replicates and statistics

Thank you for the suggestion, however all the figure legends already include the numbers of replicates and the corresponding statistical details. We are not aware of any missing information or a figure where these details are absent.

- No WT comparison in 1C so it makes comparisons difficult.

All data are compared to the WT; in Fig. 1C these are the two first grey columns. The wild type is shown in all figures.

- What are the gray boxes in extended figure 1b?

We are sorry for not making it clear. The gray boxes in this figure (now Extended Data Fig. 6a) are missing values from quadruplicate measurements. We indicated this in the previous figure legend by saying “Missing values are shown in grey”, and now by adding “Missing values from quadruplicate experiments are indicated in grey”.

- 2d and extended 1d have some karyotypes but different values. These panels weren't clear. What are they looking at?

Fig. 2d and Extended Data Fig. 1d (now Extended Data Fig. 6b) are the same cells that differ with the respect to treatment, namely either treated with Bafilomycin A1 to inhibit autophagy, or with vehicle (DMSO). This was and is-indicated in the text and in the figure legends. This experiment was performed as a control, when we wanted to find out whether inhibition of autophagy to stabilize the levels of p62 will cause an increase in the number of proteins interacting with p62 in the cytosol.

- 4d not labeled as panel d (missing label?)

We thank the reviewer for pointing this out, we have corrected this error.

- They mention SOD1 import into mitochondria in the text, but then show SOD2 in extended data. Was SOD1 a typo? The rationale for SOD1 there wasn't clear as it's mostly a cytosolic protein.

We thank the reviewer for pointing this out, we have corrected this typographical error. SOD2 is the mitochondrial matrix superoxide protein we imported in our experiment.

- Extended figure 2c was cited in the third paragraph of the results section, but it's not clear to this reviewer what this panel has to do with the text at that point. Did the authors mean to cite another panel?

This panel (Extended Data Fig. 2c, now Extended Data Fig. 2d) is correctly cited. It describes the distribution of the mitochondrial p62 proximal proteins with respect to their sub-compartmental localization, as already published in the MitoCarta 3.0 inventory.

- They didn't see extrachromosome-encoded proteins associating with p62. Can they at least confirm that these proteins are overexpressed? It's also possible that MS missed them or that they're in the p62 condensates but not abundant enough or interacting strong enough to be pulled out. Their premise is that the extra protein load causes p62 condensate build-up so this would be important.

We have created the used polysomic cell lines and shown in several previous publications that the genes on the extra chromosomes are expressed at increased levels in global transcriptome and in global proteome (e.g., Stingle et al 2012, *Mol Syst Biol*, Donnelly et al, 2016, *EMBO J*, Vigano et al, 2018*MBoC*, and more). This information was also included in the first submission manuscript in the supplementary tables providing all the proteomic data. We now added Supplementary Fig. 1a depicting the abundance changes. Note that the protein abundance increase is lower than it would be expected from the gene dosage; this so-called dosage compensation on protein level is typical for aneuploid cells (e.g., Schukken et Sheltzer, 2022).

While some proteins encoded on the extra chromosomes indeed associate with p62 (e.g., PELO and PARP4), the gene ontology enrichment analysis did not reveal significant overrepresentation of the proteins encoded on the aneuploid chromosomes among the interactors (previous Fig. 3c, now Fig. 2c). Our hypothesis is that the extra protein load overwhelms the proteostasis network, causing proteotoxic stress, protein folding defects (see our previous publication Donnelly et al, *EMBO J*, 2012), and accumulation of the p62 foci (Stingle et al, 2012, *MolSysBiol*). Importantly, the arising proteotoxic stress affects the proteome globally, and the accumulation of p62-positive aggregates containing proteins from the global proteome and enriched for the mitochondrial proteins increases with the increasing level of proteotoxic stress.

Reviewer #2 (Remarks to the Author):

In this manuscript, Amponsah et al demonstrated that proteotoxic stress in polysomy cells could compromise mitochondrial functions by sequestration of mitochondrial precursor proteins in cytosolic p62-bodies. More specifically, the authors developed unique polysomic models by chromosome transfer, which showed cytosolic p62-bodies abundantly. In addition, mitochondrial proteins were enriched within the cytosolic p62 interactome and proxitome in the polysomic cells, suggesting

mitochondrial proteins would be trapped in p62-bodies to inhibit trafficking of these proteins into mitochondria. Finally, the authors demonstrated that mitochondrial functions were impaired in the polysomic, showing reduced oxygen consumption or mitochondria DNA abundance. Protein chemistries to identify mitochondrial proteins in p62-bodies were great. These findings should provide new insights on aneuploid biology, filling a functional gap between proteotoxic stress and metabolic stress. However, there are couples of major issues to be cleared: 1) how and why p62 bodies were profoundly formed in the polysomic cells? 2) Were mitochondria in polysomic cells physiologically damaged? 3) How does gain of chromosome(s) affect metabolic networks in the polysomic models. Major concerns are concretely described below.

We are grateful to the reviewer for the interest in our work and for taking a critical look at our manuscript. We believe that better explanation and additional data will reliably address the criticisms.

Major concerns

(1) Effect of aneuploid-mediated proteotoxic stress on p62-body in polysomic cells

While the authors showed increased p62-bodies in the polysomic cells, it is unclear whether aggregated proteins were accumulated in the polysomic cells to trigger p62-body formation.

We thank the reviewer for this very relevant suggestion. In the revised version, we stained the aggresomes using a commercially available kit (Abcam, #ab139486), and confirmed that indeed aggregated proteins/aggresomes accumulate in polysomic cell lines, and that the amount of aggregates correlates with the degree of aneuploidy. These new results are now included in the Extended View Fig. 1a-c. Additionally, we show that overexpression of chaperones reduces taggregate formation in aneuploid cells (new Fig.6i-j). What exactly triggers the formation of p62 bodies in polysomic cells remains only partially understood and remains a topic of future studies. Similar observations have been already previously published by our and other labs and are duly cited in the manuscript. Protein aggregates accumulate in response to aneuploidy in yeast and mammals, most likely due to imbalance in protein complex stoichiometry (Torres et al. 2007, *Science*; Oromendia et al. 2012, *Genes Dev*; Tang et al. 2011, *Cell*; Stinglele et al. 2012, *Mol Syst Biol*; Santaguida et al. 2015, *Genes Dev*; Brennan et al. 2019, *Genes Dev*). We previously showed that ubiquitin-positive puncta increase in our polysomic cells and localize with p62 (Stinglele et al. 2012, *Mol Syst Biol*). The trigger of the p62-body formation (and aggregation) is a subject of extensive investigation in our research group.

Proteotoxic stresses, such as activation of UPR signaling for ER stress, need to be determined in the polysomic cell lines compared to their parental cells.

These results are already available in the previous publications from our and other laboratories. We described the results in the introduction (third paragraph) and cited the relevant references in the manuscript (e.g., Donnelly et al., 2014 *EMBO J*, Oromendia et al, 2012 *Gen Dev*, Zhu et al. 2019 *Science*, etc., Brennan et al, *G&D*, 2019). These papers clearly show that the cells with extra chromosomes suffer from defects in protein folding and have increased sensitivity to inhibitors of protein folding, increased dependency on protein degradation pathways, and accumulate cytosolic protein deposits. UPR signaling is not significantly upregulated, and heat shock response is also not activated in these constitutively trisomic and tetrasomic cells. Many of these previously published discoveries were obtained in our laboratory using our polysomic cell lines. Thus, we do not see any added value in repeating the experiments.

(2) Effects of overexpression of chaperon protein on p62-body

The authors showed that overexpression of chaperon proteins reduced numbers of p62-bodies in the polysomic cells, while the size of p62-bodies turned to be larger chunks (Fig. 6i), suggesting that aggregated proteins may be more accumulated in these cells. Reversely, if overexpression of chaperon proteins cleared the protein aggregation, aggregated proteins should not promote p62-body formation in the polysomic cells. The authors need to explain how overexpression of chaperon proteins affected aggregated proteins and larger chunks of p62 bodies. Rescue effects of overexpression of chaperon proteins on mitochondria activities may be determined as well.

Indeed, we observed decreased numbers but increased size of the p62-positive bodies in the polysomic cells transiently overexpressing various chaperones. Prompted by this remark, we now analyzed accumulation of aggregates in aneuploid cells upon overexpression of chaperones. This experiment showed a clear reduction of aggregates in all the analyzed cell lines. The results are now in the new Fig. 6i-k.

Like this reviewer, we were initially puzzled by the increased size of the p62 foci. Importantly, this observation is in line with several other publications that analyzed the toxicity of protein aggregates and the mechanism of chaperone activity (Mannini et al. 2012 PNAS; Emin et al. 2022 Nat. Commun.). The existing data suggests that many small aggregates in a cell are more toxic than a few big aggregates, since they have a higher propensity to interact with and disrupt membrane-bound organelles such as mitochondria (reviewed in Rinauro et al. 2024 Mol. Neurodegener.). Chaperones have been shown to consolidate many small toxic protein aggregates into larger chunks to prevent their interaction with cellular membranes, thereby decreasing their toxicity (Mannini et al. 2012 PNAS). In a related studies in yeast, it was shown that the overexpression of the mitochondrial import factor MIA40 plays an important role in decreasing the toxicity of Huntington polyQ aggregates, partially by merging many dispersed small aggregates into few huge ones (Schlagowski et al. 2021 EMBO J). Thus, the increased size and decreased number of p62-positive bodies in polysomic cells overexpressing chaperones (previous Fig. 6, now Extended Fig. 8) should be interpreted as a consolidation of many 'small' aggregates into the few bigger ones, instead of accumulation of more aggregated proteins. Thus, the reduced number of p62-positive foci despite their increased foci size might be beneficial to the cells. This also leads to a decreased p62-mitochondria colocalization, especially in the 3/3 cell line, where the proteotoxic stress and mitochondrial phenotype is less severe (previous Fig. 6, now Extended Fig. 8d-f). We now expanded the explanation of this phenomenon.

(3) Mitochondria morphology in the polysomic cells

The authors showed that polysomic cells reduced import of mitochondria proteins and accumulated mitochondria proteins in the cytosol, suggesting that mitochondria in these cells may be aberrant due to lack of their components. However, it is unclear whether mitochondria in the polysomic cells are physiologically abnormal. The authors should show the morphological changes of mitochondria, such as damaged mitochondria, in the polysomic cells. Electric microscopy may be helpful.

We respectfully disagree that the altered mitochondrial phenotype in the polysomic cells is not clear. The mitochondria in all polysomic cells show morphological defects (Fig. 5d-e), and delayed protein import is observed in all three tested cell lines (previous Fig. 7 a-e, now Fig. 7b-d), despite an uncompromised

membrane potential (previous Extended Data Fig. 5c, now Extended Data Fig. 7c). We also note that the levels of some OxPHOS proteins decrease (previous Fig. 5g-h, now Fig. 5g-l, previous Extended Data Fig. 5f, now Extended Data Fig. 7f), possibly through a stress response mechanism (Liu et. al 2019 *MBoc.*; Coyne and Chen 2018 *FEBS Lett.*), along with the elevated ROS levels (now Extended Data Fig. 7g). The mitochondrial defects in the polysomic cells are mild, which is to be expected, as the cells are viable. Using electron microscopy, while beyond the scope of the current work, might bring further insight into mitochondrial functionality in future.

Also is the expression of mitochondria-associated proteins comparable in parental and polysomic cells?

The data in the Fig. 7b (previous Fig. 7a) suggest that the levels of at least the selected mitochondrial proteins seem comparable in all cell lines, with the polysomic cells showing more accumulation of the precursor proteins. Previously published transcriptome analyses of a panel of our polysomic cell lines revealed only a marginal decrease in the levels of transcripts associated with oxidative metabolism in the 3/3 and 5/4 cell lines (Dürubaum et al. 2014 *BMC Genom.*). Similarly, there are small variable (cell line-specific) changes on the proteome level observed (Stingele et al, 2012, *Mol Sys Biol*; Yim et al, 2019, *NAR*). We now added a new figure comparing the global expression to the revised manuscript; there are no uniform significant changes between the wild type and aneuploids (Supplementary Fig. 1f).

(4) Effects of chromosome gain on metabolic networks

The authors showed polysomic cells compromised mitochondria function by sequestration of mitochondria proteins in p62-bodies. Mitochondria dysfunction may change metabolic networks in the polysomic cells presumably to shift TCA cycle to glycolysis. The authors should show the effects of mitochondria disfunction on the metabolic networks in the polysomic cells.

This is an excellent point. In fact, it has been previously shown in one of the early publications on aneuploidy by the group of Angelika Amon that the lactate production (and several other metabolic features related to TCA/glycolysis) were altered in trisomic murine cells, suggesting metabolic remodeling consistent with mitochondrial impairment (Williams et al, 2008, *Science*). While further analysis is beyond the scope of this manuscript, in future, we plan to analyze this aspect in detail, as we believe that the shift from TCA to glycolysis might be an adaptation to aneuploidy-induced mitochondrial stress and might explain some aspects of cancer biology, such as the frequent Warburg effect in tumors. In the revised manuscript, we discuss in more details the previous findings, and how our observations fit with these findings.

(5) Aneuploidy-related p62 protein expression and autophagy flux

Fig1b: p62 protein expression changes in polysomic cells are shown.

However, no data show the autophagy flux state; please show the flux for LC3 (I and II) (on the same membrane, if possible).

This data has been published already in the two previous publications from our laboratory, which are cited in the manuscript (Stingele et al. 2012 *Mol Syst Biol*; Krivega et al. 2021 *Commun Biol*). There is no defect in autophagy flux as observed by LC3I and LC3II immunoblotting, as well as by double-tagged LC3 (with GFP and RFP).

(6) Nuclear localization of p62 in the polysomic cells
HADHB (mitochondrial hydroxy acyl-CoA dehydrogenase trifunctional multienzyme complex subunits) and p62 showed nuclear localization in polysomic cells. Do proteins co-localizing with nuclear p62 exhibit normal autophagic flux?

We respectfully disagree with the interpretation that the image showing p62 and HADHB colocalization in the 5/4 cell line (previously Fig. 2c, now Extended data Fig. 5b) demonstrates nuclear localization. This is a 2-D reconstruction from a 3-D image and thus, such impression can arise upon collapsing the Z-stacks. As can be seen in the other cells from the same image, this indicates perinuclear clustering. For the avoidance of any doubt, we have attached the full microscopy image from which this image was cropped. As seen in the adjoining cells, p62 and HADHB do not localize to the nucleus (Fig. R2). To avoid possible misinterpretations, we now used better representative images in the revised version.

Fig. R2: Microscopy images showing p62-HADHB colocalization in 5/4 cells in Fig. 2c. a, Full microscopy image from which image in Fig. 2c was cropped. **b,** Replacement image for Fig. 2c and **c,** Existing image in Fig. 2c. showing all channels. Note that this is currently an Extended data Fig. 5b.

The authors should confirm normal autophagy flux by an autophagy flux assay with a fluorescent probe such as "tandem fluorescent protein-tagged LC3" or "GFP-LC3-RFP-LC3ΔG".

Thank you for the suggestion, however, as already explained above, the autophagy flux in polysomic cells has been previously determined by us in these cell lines, using the tandem-tagged LC3. The autophagy flux is functional and comparable with the parental cell line (Stingele et al 2012 *Mol Syst Biol*; Krivega et al. 2021 *Commun Biol*). We cited these references and we do not see any added value of repeating these experiments.

The authors also need to compare and quantify the co-localization rate of p62 on autophagosomes, which may be functionally intact with the cytoplasmic localization.

We thank the reviewer for this suggestion. We now quantified the colocalization of LC3b with p62 in the parental WT cell line and in three different aneuploid cell lines. There is no uniform effect, and no significant increase of the colocalization compared to the wild type. These new results are now the part of the Extended Data Fig. 4d-e.

Other Concerns

Fig2C, Extended Data Fig1: The expression level of mitochondria-associated proteins appears to decrease. On the other hand, in Fig. 7A, the expression levels of HADHA and HADHB are unchanged (precursor is increased).

We thank the reviewer for pointing this issue out. The previous Fig. 2c (now Extended Data Fig. 5b) shows representative confocal images of p62 colocalization with selected interactors in WT and 5/4 cells, including HADHB, while the data Fig. 7a (now Fig. 7b) shows levels of precursor and mature forms of selected mitochondrial proteins, including HADHB, by immunoblotting.

We do not think that the levels of HADHB protein are lower in the polysomic cells compared to the WT cells based on the microscopy images. As seen below in the unprocessed fluorescence images from individual channels, the HADHB staining is more diffused in the WT, while the signal is more concentrated in foci-like structures in the polysomic cell line (Fig. R3). The intensity of the signal appears similar in the 5/4 cells. We suggest that while the abundance is comparable in WT and polysomic cells, the HADHB becomes increasingly sequestered in p62 foci, thus, less diffusely distributed.

We would also like to point out that in general immunoblotting is more suitable for quantitative analysis of protein abundance and the data from these two different methods might be difficult to compare.

Fig2C: Present a Fig for each fluorescence channel, showing the change in fluorescence intensity of HADHB.

Fig. R3. As requested, we show here the unprocessed fluorescence images for all the channels, using WT and 5/4 as an example. Please see the explanation above. Additionally, we now exchanged the representative figure (now Extended Figure 5b).

Fig2C, Extended Data Fig1: Do HADHB and TOMM20 expression co-localize?

Both HADHB and TOMM20 are annotated mitochondrial proteins localized to the inner and outer membranes, respectively. It is just intuitive that they will likely partly colocalize in human cells. We do not understand added value of this experiment.

Fig4:

>Surprisingly, we found no over-representation of mitochondrial proteins in the polysomy-specific p62 cargo in autophagosomal lumen (Fig. 4c, d, Extended Data Fig. 3c).

There is no "d" in Fig. 4.

The reviewer is right, and we are grateful for pointing this out. This was an oversight. The panel to the right is "d". We corrected the labeling in the revised manuscript.

Reviewer #3 (Remarks to the Author):

Aneuploidy affects cellular physiology and development. Previous studies suggested that increased expression of surplus proteins encoded on the extra chromosomes may contribute to the abnormalities. This results in proteostatic crisis as manifested by increased recruitment of aggregated proteins to p62, a key player in aggrephagy and other types of autophagic activities. However, it remains unclear whether this affects the homeostasis of specific groups of proteins, followed by specific cellular defects in aneuploidy cells. In the current manuscript, the authors addressed this gap of knowledge.

The authors elegantly generated cell lines that contain extra copy (or copies) of chromosomes 3, 5, 13 and 21. They found that the formation of p62 foci correlates with the loading of surplus protein coding

genes in these polysomic cells. Proteomic analysis of p62 interactors using immunoprecipitation and proximity labeling assay showed enrichment of many mitochondrial proteins, but not those encoded on the extra chromosomes in p62 bodies in the cytosol. The authors also showed that cells with extra chromosomes contain stressed and seemingly less functional mitochondria, with reduced levels of mitochondrial proteins and mtDNA. Interestingly, the polysomic cells appear to accumulate unimported mitochondrial precursor proteins. In organello protein import assay also showed that The overall mitochondrial protein import capacity is also reduced in polysomic cells. Based on these observations, it was concluded that sequestration of mitochondrial proteins by cytosolic p62 bodies may impairs mitochondrial health. This may contribute to cellular stress in polysomic cells.

This is an interesting study as it uncovered the mitochondrial protein import process as a vulnerable process affected by polysomy. The authors provided compelling evidence that mitochondrial precursors are sequestered by p62 in aneuploidy cells, supported by solid data from two independent proteomic approaches. This is an important step towards the better understanding of how aneuploidy affects cellular proteostasis. Strong data also comes from the experiments showing that mitochondrial protein import is affected and that unimported mitochondrial precursor proteins accumulate in polysomic cells. There are a few weaknesses that damper the overall enthusiasm which need to be addressed.

We are grateful to the reviewer for the interest in our work, the accurate description of our results and for taking a critical look at our manuscript. We believe that better explanation and additional data will sufficiently address the criticisms.

Major weaknesses:

1. One of the key observations is that the formation of p62 foci correlates with the loading of surplus protein coding genes in these polysomic cells. As noted by the authors, the p62 gene is present on chromosome 5. This unfortunately confounds the quantification of p62 bodies in the 5/3 and 5/4 cell lines, as the increase of p62 bodies may also result directly from elevated p62 copy number. The data in the 13/3 cell line is convincing, showing the sequestration of mitochondrial proteins. Perhaps, a negative control cell line with increased p62 expression may be used to validate the data in 5/4 cells.

We understand the reviewers' criticism and appreciate this excellent suggestion. As recommended by the reviewer (and also by the reviewer #1), we added a control where we overexpressed p62 in the parental cell line to a comparable degree: a 5-fold increase compared to the parental cell line in the 5/4 cell lines, and 7-fold increase in the WT cell line overexpressing p62 (now Fig. 4c-d). p62-IP in these cell lines showed lower enrichment of mitochondrial proteins in the overexpressing diploid cell line than in the analyzed polysomic cell lines (3/3 and 5/4 cell lines) (now Fig. 4f). This results, now shown in the Figure 4c-f, strengthen our hypothesis that p62-positive cytosolic bodies sequester mitochondrial proteins specifically in the cells with extra chromosomes, and the overexpression of p62 alone is not sufficient to result in the comparable phenotype.

2. The authors showed changes to mitochondrial localization, reduced mtDNA copy number and OXPHOS components in polysomic cells (Fig. 5). This could just reflect a general reduction of mitochondrial biogenesis in response to global proteostatic stress, instead of a specific functional impairment and perturbed mitochondrial genome maintenance as claimed in the manuscript.

We fully agree with the reviewer on this point. Our data indicate that the gain of even a single chromosome causes proteotoxic stress and that this alters mitochondrial homeostasis. We do not intend to push our conclusions towards “a specific functional impairment” and do not claim that the observed mitochondrial phenotypes are due to the perturbed mitochondrial genome maintenance. Rather, we investigate the mechanism by which proteotoxic stress of aneuploid cells can affect mitochondria. Our conclusion is that the impaired protein folding leads to the accumulation of aggregated proteins, which sequester mitochondria precursor proteins, and thereby impair mitochondrial function.

On the other hand, the data with mitochondrial functional assay are problematic. First, no details were provided regarding how the experiments were executed, except that a reference is provided in which isolated mitochondria from yeast cells were analyzed.

Indeed, we did not include a description of the oxygen consumption assay in the methods section of the manuscript. We apologize for this oversight and are very grateful to the reviewer for pointing this out.

How much isolated mitochondria are used in the assays? I am afraid that the scale on the Y axis may be wrong (Extended Data Fig. 6b).

We thank the reviewers for pointing this out. 100 µg of mitochondria were used per reaction, and we now added this information to the Methods section. We thank the reviewer also for noticing the Y axis labeling. We corrected it, and the scale on the y-axis is now in the negative (now Extended Data Fig. 7k).

It is unclear why NADH is used as a substrate, instead of using the standard substrates glutamate/malate. Mammalian mitochondria do not seem to directly oxidize NADH from the cytosolic side. Controls for standard assays are missing, including oligomycin-inhibited and FCCP-stimulated respiration, making it difficult to evaluate whether the isolated mitochondria are intact.

The import experiments were carried out in the presence of cytosolic extracts of reticulocyte cells (which included the radiolabeled proteins). These lysates contained the cytosolic enzymes required to efficiently shuttle electrons from NADH into mitochondria by the malate-aspartate shuttle reflecting the physiological conditions in mammalian cells. Moreover, we would like to point out that the mitochondria we used for the oxygen consumption assay were isolated similarly to (and sometimes from the same preparation as) those used in the *in-organello* import assay. These mitochondria were immediately used after purification without freezing. Without intact isolated mitochondria, the import would be impossible. In addition, we performed control experiments where we treated mitochondria isolated from the parental cells briefly with CCCP to collapse the membrane potential before the oxygen consumption assay. These results were shown in Extended Data Fig. 6b-c (now in the Extended Data Fig. 7 k, l). Additionally, we expanded the method description to explain this aspect.

The data on Opa1 processing and Drp1 levels provide little information regarding mitochondrial functionality (Extended Figure 6d-e). It seems to me that the authors try to push the idea that sequestration of mitochondrial precursor proteins by p62 causes mitochondrial dysfunction. This would be fine given the drastic reduction of OXPHOS components in the polysomic cells, as long as the experiments should be done correctly. But on the other hand, mitochondrial bioenergetics is very flexible in cell lines. Mitochondrial protein import stress can affect cell physiology in many other ways, without necessarily affecting respiration in a major way.

We agree with the reviewer on this point. We aimed to demonstrate that the gain of even a single chromosome causes proteotoxic stress and alters mitochondrial homeostasis, including mitochondrial dynamics. The result on Opa1 processing and Drp1 levels partly support this (previous Extended Data Fig. 6d). However, we agree that this data is not providing any insight into mitochondrial functionality, and therefore we removed this figure from the revised version. The main observed changes are the altered mitochondrial network, reduced levels of mtDNA and mitochondrial proteins, and impaired protein import. Additionally, we now measured ROS levels in two polysomic cell lines, revealing that ROS are increased in these cells. This new data provides further evidence of reduced mitochondrial function (now Extended data Fig. 7g).

3. Figure 6 – The authors measured p62 body density and size in polysomic cells overexpressing several chaperones and heat shock transcriptional factor. Based on the reduction of p62 foci number, the authors concluded that “...the proteotoxic stress and mitochondrial defect can be rescued by overexpression of cytosolic chaperones...”. And that “Our data suggests that impaired protein folding and chronic overexpression of extra proteins in polysomic cells facilitate formation of cytosolic p62 bodies that sequester mitochondrial proteins, thus impairing mitochondrial functions”. These conclusions are poorly supported by the data. The p62 foci are clearly larger in cells overexpressing the chaperones. This may also be interpreted as increased aggregation propensity, instead of alleviating proteostatic stress. It may be an oversimplification in that expressing one single chaperone can fix the problem, especially no supporting data are available showing that mitochondrial dysfunction is rescued by the chaperones.

To further elucidate this aspect, we now (1) added another trisomic cell line to these experiments, the results of which are comparable with the other two polysomies, and (2) tested the formation of aggregates in the cells overexpressing chaperones and HSF1. This latter experiment clearly showed reduced aggregate formation upon overexpression. This new data strongly supports our previous conclusion and is now shown in the Fig. 6i-k. However, we agree with the reviewer that the interpretation needs to be made with a caution, and thus we toned down the conclusions.

Given that mitochondrial protein overexpression and unbalanced protein loading/import are known to directly cause proteostatic stress in the cytosol, it is possible that disturbed cytosolic proteostasis by polysomy may directly disrupt mitochondrial precursor delivery to mitochondria and the formation of p62 foci may just be a stress response in the cytosol. The authors need to discuss this in the manuscript.

We agree with the reviewer that it remains to be understood what is the cause and what is the consequence. In the revised manuscript, we amended the statements to reflect this fact:

Last Results chapter: “We propose that cytosolic proteotoxic stress due to defective protein folding is associated with mild, but chronic reduction of mitochondrial protein import and impaired mitochondrial function in aneuploid cells. The unimported mitochondrial precursor proteins accumulate in the cytosol, where they get sequestered into p62-positive bodies.”

Discussion: “Currently, it remains unclear what exactly is the trigger of the formation of the p62-positive cytosolic bodies enriched for mitochondrial proteins. On one hand, disturbed cytosolic proteostasis in polysomic cells may impair mitochondrial precursor delivery to mitochondria, which leads to formation

of p62 bodies. Alternatively, proteostasis impairment may lead to formation of the p62 bodies, which have high propensity to interact with metastable mitochondrial precursor proteins. The specific molecular mechanisms as well as other aspects, such as contribution of oxidative stress, or changes in lipid metabolism, are subject of ongoing research.”

The reasons for the increased accumulation of cytosolic p62 bodies in aneuploid cells remain unclear.

4. It is also misleading when stating that “.....the impaired protein folding and imbalanced protein homeostasis due to aneuploidy initiates protein aggregation in cytosolic p62 bodies, which then sequester mitochondrial proteins before they can be imported into mitochondria”. p62 normally sequesters ubiquitinated proteins in aggregates. It is unlikely that these proteins readily regain the unfolded state for further import.

We agree with the reviewer, in fact, this is what we propose. We never wished to propose that the proteins may regain their unfolded status and become available for import. We now amended the sentence to avoid any misunderstandings.

Minor comments:

1. Figure 1 - Figure 1c, which chromosome encodes beta-actin should be indicated in the legend.

Thank you, corrected.

In the text - “.....demonstrating normal autophagic flux...”. As no rate of flux is measured, “normal” is unjustified.

As already indicated, these results have been previously published by us, and are referenced in the manuscript (Stingele et al 2012 *Mol Syst Biol*; Krivega et al. 2021 *Commun Biol*). We emphasize this fact better in the revised version.

Figure 1D – The images lack the delineation of cell boundary, making it difficult to ascertain whether the green puncta come from a single cell.

We used the ImageJ image contrast and brightness tool to manually delineate cell boundaries before computationally performing quantification of the green p62 puncta. The data is indicated per cell surface area, as described in the Online methods section of the manuscript.

Figure 1h – 1i, add p62 to the Y axis.

Thank you, corrected.

2. Figure 3d – please check whether the p values are correct.

We thank the reviewer for pointing this error out. The negative exponentials are missing. We will correct it.

3. Extended Data Fig. 6b – “5/4” instead of “54”

We thank the reviewer for pointing this error out. We will correct it.

4. The authors may consider moving Extended data Fig. 1b to the main figure, given the importance of the data for supporting the overall conclusion.

We thank the reviewer for this suggestion, but we now completely restructured the manuscript, and this figure is now Extended Data Fig. 4.

5. Figure 7b-d – what does “20%” denote?

20% of the *in-silico* synthesized substrate (precursor) added to the isolated mitochondria for the import assay. We will indicate it in the figure legend.

6. Ref. 37 is incomplete.

Thank you, corrected.

7. Page 11, line 15: broken sentence.

Thank you, corrected.

8. Page 11, “mitochondrial internal membrane-localized”. I guess it meant “...inner membrane...”.

Thank you, corrected.

9. Page 11, line 15, broken sentence.

Thank you, corrected.

Reviewer #4 (Remarks to the Author):

"I co-reviewed this manuscript with one of the reviewers who provided the listed reports. This is part of the Nature Communications initiative to facilitate training in peer review and to provide appropriate recognition for Early Career Researchers who co-review manuscripts."

We thank the reviewer for co-reviewing our manuscript.